# BEYOND SPECTRA: EIGENVECTOR OVERLAPS IN LOSS GEOMETRY

**Gabriel C. Mel**[1,2]
[1] Centre de Recerca Matemàtica, Bellaterra (Barcelona), Spain
[2] CRG (Barcelona Collaboratorium for Modelling and Predictive Biology),
Dr. Aiguader 88, Barcelona 08003, Spain
`meldefon@gmail.com`

## ABSTRACT

Local loss geometry in machine learning is inherently a two-operator concept. While a single loss is locally characterized by its Hessian spectrum, practical learning depends on both training and test losses, whose joint geometry is determined not only by their spectra but by the alignment of their eigenspaces. We establish general foundations for this two-loss geometry by deriving a universal local fluctuation law: the expected test-loss increment under small training perturbations is a trace combining train and test spectral data with a precise factor quantifying eigenvector overlap. We further prove a transfer law describing how overlaps transform under noise. As a solvable model, we apply these results to ridge regression under arbitrary covariate shift, where operator-valued free probability yields asymptotically exact overlap decompositions that identify overlaps as the natural quantities for specifying shift, and resolve multiple descent: error peaks are governed by eigenspace misalignment rather than Hessian ill-conditioning alone. We then validate the fluctuation law in multilayer perceptrons, develop scalable estimators for overlap functionals based on subspace iteration and kernel polynomial methods, and apply them to a ResNet-20 trained on CIFAR-10, showing that class imbalance reshapes train–test geometry through induced misalignment. Together, these results establish eigenvector overlaps as the fundamental missing ingredient in local loss geometry, providing both theoretical foundations and practical tools for analyzing generalization in modern neural networks.

## 1 INTRODUCTION

Modern learning algorithms are inherently local, and sources of randomness (stochastic gradients, finite-sample variability, and distributional drift) are often small relative to the underlying signal. A local quadratic approximation to the loss thus provides a natural setting for analyzing learning. When the focus is a single loss, local geometry is fully captured by the Hessian spectrum. Crucially, however, machine learning involves (at least) *two* losses—train and test—and so local loss geometry involves two quadratic approximations. Their joint geometry is not captured by Hessian spectra alone; it requires a critical additional ingredient: *eigenvector alignment*, or overlaps.

Despite the fundamental importance of eigenvector overlaps, most studies to date have centered on Hessian eigenvalue distributions—often explicitly equating spectra with loss geometry. The literature is extensive and examines Hessians from several complementary angles, including: (i) empirical measurement of eigenvalue distributions and their training-time evolution, with links to optimization stability (Sagun et al., 2017; Ghorbani et al., 2019; Yao et al., 2019); (ii) random-matrix-theoretic and mean field models (Pennington & Bahri, 2017; Pennington & Worah, 2018; 2019; Liao & Mahoney, 2021; Karakida et al., 2019); (iii) class- and layer-structured spectral phenomena, such as identifiable outliers tied to data and architecture (Papyan, 2020; Sankar et al., 2021); and (iv) Hessian/Fisher analyses relating sharpness (as measured by eigenvalue magnitude) to stability and generalization (Keskar et al., 2017; Cohen et al., 2021; Yao et al., 2019). These studies give fundamental insight into aspects of loss curvature, but ignore directional information that becomes relevant as soon as one compares two operators.

The need to go beyond spectra is well understood in random matrix theory, where eigenvector consistency and overlaps with population directions are central objects in spiked models and correlated ensembles (Johnstone, 2001; Paul, 2007; Nadler, 2008; Benaych-Georges & Nadakuditi, 2011; Bun et al., 2017; Landau et al., 2023). There, eigenvalues alone do not determine statistical performance; rather, risk depends on how sample and population eigenvectors align. Related phenomena have also been observed in machine learning, where eigenspace overlap has been used to characterize shared Hessian structure across independently trained networks (Wu et al., 2022) and to predict compressed embedding performance (May et al., 2019). Here, we adapt this principle to learning-theoretic questions by analyzing train-test alignment. We show that overlap measures between the training fluctuations covariance (intimately related to the training Hessian; see Results) and the test Hessian yield a decomposition of generalization error into spectral and alignment components.

Applying this perspective to ridge regression resolves the puzzle of anisotropy-induced multiple descent (see Chen et al. (2021); Li & Wei (2021); Mel & Ganguli (2021); Meng et al. (2023) for several distinct forms of multiple descent). Existing high-dimensional analyses connect interpolation peaks to eigenvalue distribution of the design matrix (Singh et al., 2022; Chen & Mei, 2022). However, in anisotropic settings where error exhibits multiple peaks despite monotonically decreasing minimum training eigenvalue, spectra alone do not explain generalization. Making overlaps explicit, we show that the appearance of multiple peaks is governed by the alignment between train and test Hessian eigenspaces. This corrects interpretations that implicitly attribute sample-wise multiple descent to spectrum ill-conditioning (Chen & Mei, 2022; Mel & Ganguli, 2021; Mel & Pennington, 2022), and yields a simple geometric picture that may prove useful for understanding more complex models.

A second arena where eigenvector orientation is essential is generalization under domain shift. In covariate shift settings, high-dimensional risk formulas in random feature models (Tripuraneni et al., 2021) can be naturally expressed in terms of train/test covariance spectra and their overlaps. More broadly, many domain generalization methods encourage cross-domain invariance by aligning gradients, Fisher information, or Hessian statistics across domains (Rame et al., 2022; Hemati et al., 2023; Le & Woo, 2024), or by imposing structural constraints such as elliptic regularization (Hasan et al., 2025). These approaches seek robustness across *many* possible unseen domains. Our perspective is complementary: rather than enforcing invariance, we derive explicit formulas for performance on a *specific* target domain. In particular, under covariate shift we show that, even with covariance spectra held fixed, varying overlap structure alone can increase or decrease test risk. Overlaps therefore provide a natural quantitative measure of the shift itself, predicting when a given domain change will help or hurt in a way that spectrum-only or domain-agnostic analyses cannot.

Loss curvature plays an important role in both classical and modern analyses of generalization. Classical asymptotic corrections such as the Takeuchi Information Criterion (TIC) express the generalization gap in terms of the local curvature of the population loss (see, eg., Thomas et al. (2020)). A second line of work uses curvature information at training time: sharpness-aware and curvature-regularized methods—including SAM (Foret et al., 2021) and its Fisher- and curvature-regularized variants (Kim et al., 2022; Wu et al., 2024)—bias optimization toward flatter regions of the training loss, motivated by the heuristic that such regions generalize better. Both perspectives are fundamentally single-loss. By contrast, our framework is explicitly two-loss: we do not assume any relationship between train and test losses. Given a training loss and perturbation model, we characterize test performance—potentially on a distinct domain—through the joint spectra of the two Hessians and their eigenvector overlap. The TIC emerges as a limiting single-loss case, but our theory reveals how alignment governs generalization beyond spectrum-based criteria.

Translating our overlap theory into practice at modern scale requires algorithms that go beyond spectral density estimation. A substantial literature has developed linear algebraic tools for implicit matrices, including polynomial/quadrature approaches and stochastic trace methods such as Hutchinson and Lanczos-based quadrature (Golub & Meurant, 2009; Lin et al., 2016; Ubaru et al., 2017). These and related techniques have been adapted to deep learning to estimate Hessian spectral densities and extremal eigenpairs efficiently (Adams et al., 2018; Papyan, 2019; Ghorbani et al., 2019; Yao et al., 2019). Building on these, we develop novel estimators for overlap functionals between pairs of Hessians (train-test, population-sample), and use them to study class imbalance effects on train–test geometry.

The resulting picture is that local geometry in machine learning is fundamentally bivariate: spectra determine the curvatures of train and test losses, while eigenvector overlaps determine how these curvatures interact to produce test error.

## 2 CONTRIBUTIONS

1. **Two-loss theory of local geometry.** We introduce a novel two-loss framework for local loss geometry that incorporates both spectra and overlaps 3.1, rectifying a widespread oversimplification that equates spectra with geometry.

2. **General foundations.** We derive and test a universal local fluctuation law showing how overlaps impact generalization 3.1.1, and a general transfer law dictating how eigenvector overlaps are transformed by noise 3.1.2.

3. **Explicit formulas for high-dimensional ridge regression.** Combining tools from random matrix theory with our overlap transfer law, we provide closed-form expressions for the overlap function between train and test Hessians in anisotropic ridge regression 3.2.

4. **Unified explanation of covariate shift and multiple descent.** We show that covariate shift is naturally quantified by eigenvector overlaps 3.2.1, and that overlaps analytically resolve the puzzle of multiple descent 3.2.2.

5. **Empirical validation in neural networks.** We confirm our theoretical predictions in multilayer perceptrons, and use overlap machinery to show that the training Hessian acts as a filter shaping optimization 3.3.

6. **Scalable algorithms for Hessian overlaps.** We develop novel, scalable numerical methods for estimating Hessian eigenvector overlaps in large-scale models, enabling practical use of our theory in modern deep learning 3.4.

7. **Train-test misalignment under class imbalance.** We show that class imbalance in CIFAR induces misalignment between train and test Hessians, explaining the effects of class imbalance in terms of train-test loss geometry. 3.4.

## 3 RESULTS

### 3.1 THEORETICAL FOUNDATIONS

Prior theoretical and empirical work frequently uses geometric descriptors of the loss landscape—such as "sharp" versus "flat" minima or valley structures—yet the relationship between these geometric notions and generalization remains imprecise. We begin by establishing general foundations for two-loss geometry to formalize this connection, and then derive a fluctuation law that characterizes how perturbations to the training loss propagate to changes in the test loss.

Let $w \in \mathbb{R}^d$ denote the $d$-dimensional parameter vector of a model $f_w$, and let $\mathcal{L}_{\text{train}}(w, \epsilon), \mathcal{L}_{\text{test}}(w)$ be the (twice-differentiable) train and test loss functions. The train loss $\mathcal{L}_{\text{train}}$ is parameterized by a small variable $\epsilon$ representing a general training perturbation. We remain agnostic about the source of the perturbation, which could be any combination of label/input noise, distributional drift, sampling effects, etc. Throughout, we write $w_0$ for the minimum of the unperturbed loss $\mathcal{L}_{\text{train}}(w, 0)$.

By analogy with the one-loss case, we refer to the pair of quadratic approximations obtained by second order expansion of $\mathcal{L}_{\text{train}}, \mathcal{L}_{\text{test}}$ around a point as the *local two-loss geometry*. Concretely, we define the perturbation gradient, and train and test Hessians as follows:

$$z := d\,\nabla_w \mathcal{L}_{\text{train}}(w_0, \epsilon), \qquad H_{\text{train}} := d\,\nabla_w^2 \mathcal{L}_{\text{train}}(w_0, \epsilon), \qquad H_{\text{test}} := d\,\nabla_w^2 \mathcal{L}_{\text{test}}(w_0), \quad (1)$$

(note the scalings, chosen for convenience) and introduce the quadratic surrogate losses:

$$\mathcal{L}_{\text{train}}^{\text{quad}}(w) = \mathcal{L}_{\text{train}}(w_0, \epsilon) + \frac{1}{d} z \cdot \Delta w + \frac{1}{2d} \Delta w^\top H_{\text{train}} \Delta w, \qquad (2)$$

$$\mathcal{L}_{\text{test}}^{\text{quad}}(w) = \mathcal{L}_{\text{test}}(w_0) + \frac{1}{d} z_{\text{test}} \cdot \Delta w + \frac{1}{2d} \Delta w^\top H_{\text{test}} \Delta w, \qquad (3)$$

where $z_{\text{test}} := d\,\nabla_w \mathcal{L}_{\text{test}}(w_0)$ is the normalized test gradient and $\Delta w := w - w_0$. Finally, we define the unperturbed test loss and test loss increment as follows:

$$\mathcal{L}_0 := \mathcal{L}_{\text{test}}^{\text{quad}}(w_0), \qquad \Delta \mathcal{L} := \mathcal{L}_{\text{test}}^{\text{quad}}(w_0 + \Delta w) - \mathcal{L}_{\text{test}}^{\text{quad}}(w_0). \qquad (4)$$

### 3.1.1 LOSS FLUCTUATIONS ARE GOVERNED BY EIGENVECTOR OVERLAPS

Generically, the effect of a perturbation is to induce a small gradient $z$ at the unperturbed minimum $w_0$, yielding a new minimum of $\mathcal{L}_{\text{train}}^{\text{quad}}$ at a displacement $\Delta w$. Directly minimizing (2) gives the perturbation-induced displacement $\Delta w = -H_{\text{train}}^{-1} z$. We sometimes refer to the perturbation gradient $z$ as the *injected noise* and to $\Delta w$ as the *(inverse-Hessian) filtered noise*. Substituting the displacement into $\mathcal{L}_{\text{test}}^{\text{quad}}$ yields the following expression for the test loss increment,

$$\Delta \mathcal{L} = -\tfrac{1}{d} z_{\text{test}}^{\top} H_{\text{train}}^{-1} z + \tfrac{1}{2d} z^{\top} H_{\text{train}}^{-1} H_{\text{test}} H_{\text{train}}^{-1} z. \tag{5}$$

Equation (5) represents the simplest model capturing the interaction of nontrivial train and test geometry in the context of noisy learning. The first-order effect is structurally simple—and, in several natural cases (e.g., label noise under MSE, analyzed below), vanishes exactly in expectation. The second order term, in contrast, involves interaction between train and test curvatures: letting $C_{\text{train}} := \mathbb{E}[\Delta w \Delta w^{\top}] = \mathbb{E}[(H_{\text{train}}^{-1} z)(H_{\text{train}}^{-1} z)^{\top}]$ be the displacement covariance, its expectation is $\tfrac{1}{2d}\text{tr}[H_{\text{test}} C_{\text{train}}]$. This simple trace expression suggests the importance of alignment between directions of large training displacement and directions of large test Hessian eigenvalue. One of the main theoretical contributions of this work is the following theorem making this intuition precise.

**Theorem 1** (Overlap local fluctuation law). *Let $\mu_{\text{test}}, \mu_{\text{train}}$ be the spectral measures of $H_{\text{test}}, C_{\text{train}}$, and define $\tfrac{1}{d} O(\lambda_1, \lambda_2)$ as the mean squared cosine angle between eigenvectors of $H_{\text{test}}, C_{\text{train}}$ at eigenvalues $\lambda_1, \lambda_2$. Assume $\mathbb{E}[\Delta w] = 0$. Then*

$$\mathbb{E}[\Delta \mathcal{L}] = \tfrac{1}{2} \iint \lambda_1 \lambda_2 O(\lambda_1, \lambda_2) \, \mu_{\text{test}}(d\lambda_1) \, \mu_{\text{train}}(d\lambda_2). \tag{6}$$

Equation (6) shows that neither train nor test spectra alone predict the expected generalization impact of noise: the decisive quantity is how training-induced displacement directions route into test-sensitive directions via the overlap kernel $O(\lambda_1, \lambda_2)$. In particular, large test error arises when when high-variance displacement directions (large $\lambda_2$, corresponding to low-curvature train directions) substantially overlap high-curvature test directions (large $\lambda_1$).

*Proof sketch.* Letting $(\lambda_i^{\text{test}}, u_i^{\text{test}}), (\lambda_j^{\text{train}}, u_j^{\text{train}})$ be the eigenvalues/eigenvectors of $H_{\text{test}}, C_{\text{train}}$,

$$\tfrac{1}{2d}\text{tr}[H_{\text{test}} C_{\text{train}}] = \tfrac{1}{2} \tfrac{1}{d^2} \sum_{i=1}^{d} \sum_{j=1}^{d} \lambda_i^{\text{test}} \lambda_j^{\text{train}} [d \, (u_i^{\text{test}} \cdot u_j^{\text{train}})^2]. \tag{7}$$

Writing the double sum as an integral over the spectral measures of $H_{\text{test}}, C_{\text{train}}$ yields (6). See Appendix B.2 for details. $\qquad\square$

While we do not treat stochastic optimization explicitly, in the same local quadratic regime, noisy gradient descent yields a curvature-filtered steady-state covariance that, when substituted for $C_{\text{train}}$, yields the same overlap fluctuation law (see Appendix B.2.2).

### 3.1.2 OVERLAP TRANSFER LAW

In many situations one must consider the overlaps between an operator $A$ and a noisy transformation of another operator $B$, written $\hat{B}$. For example, below in the context of ridge regression with anisotropic gaussian inputs, we consider the case that $A, B$ correspond to the population test and train covariances, while $\hat{B}$ is the *sample* train covariance. More generally, $A, B$ could represent the population test and train Hessians, and $\hat{B}$ the empirical train Hessian. In such cases, one needs a way of combining the population overlaps $O_{A,B}$ with the noise, specified by $O_{B,\hat{B}}$. We prove the following appealing transfer law in Appendix B.3:

**Theorem 2** (Free transfer law for overlap functions). *Let $\hat{B} = F(B, X)$ be a matrix rational expression. If $X$ is free from $A, B$, then*

$$O_{A,\hat{B}}(a, \hat{b}) = \int O_{A,B}(a, b) O_{B,\hat{B}}(b, \hat{b}) \, \mu_B(db). \tag{8}$$

(Freeness is a notion of independence that is suited to large random matrices and holds asymptotically for a wide range of common random matrix models; see Appendix B.3.) Theorem 2 entails a simple overlap calculus that can be used to compute overlap functions of complex matrix models from simpler ones. In Appendix C, we use (8) to quickly derive expressions for train-test Hessian overlap functions in anisotropic ridge regression.

## 3.2 HESSIAN OVERLAPS GOVERN GENERALIZATION IN LINEAR REGRESSION

We now consider ridge regression, where the preceding theory is exact. Let training inputs $x \in \mathbb{R}^d$ have covariance $\Sigma_{\text{train}} := \mathbb{E}[xx^\top]$, and assume linear output with Gaussian label noise:

$$y(x) = \tfrac{1}{\sqrt{d}} w_*^\top x + \xi, \qquad \xi \sim \mathcal{N}(0, \sigma^2).$$

We will also assume for convenience that $w_* \sim \mathcal{N}(0, I)$, so that the signal to noise ratio is $\bar{\text{tr}}\, \Sigma_{\text{train}}/(\bar{\text{tr}}\, \Sigma_{\text{train}} + \sigma^2)$ ($\bar{\text{tr}}$ denotes the dimension normalized trace). Given a training set consisting of $X \in \mathbb{R}^{m \times d}$ (rows $x^\top$) and labels $y \in \mathbb{R}^m$, ridge regression chooses $w \in \mathbb{R}^d$ to minimize

$$\mathcal{L}_{\text{train}}(w) = \tfrac{1}{2m} \big\| y - \tfrac{1}{\sqrt{d}} Xw \big\|^2 + \tfrac{\lambda}{2d} \|w\|^2, \qquad \lambda \geq 0. \tag{9}$$

We write $\alpha := m/d$ for the sampling ratio. The (excess) test loss is measured with test inputs with (possibly different) covariance $\Sigma_{\text{test}}$:

$$\mathcal{L}_{\text{test}}(w) := \tfrac{1}{2} \mathbb{E}_{x,\xi}\Big[\big(\tfrac{1}{\sqrt{d}} w^\top x - y(x)\big)^2\Big] - \tfrac{1}{2}\sigma^2 = \tfrac{1}{2d}(w - w_*)^\top \Sigma_{\text{test}} (w - w_*).$$

With the scalings of 3.1, one has $H_{\text{train}} = \hat{\Sigma}_{\text{train}} + \lambda I$ and $H_{\text{test}} = \Sigma_{\text{test}}$, where $\hat{\Sigma}_{\text{train}} := X^\top X / m$ is the training set sample covariance. Note $\hat{\Sigma}_{\text{train}} \to \Sigma_{\text{train}}$ for large $\alpha = m/d$.

We now apply the local fluctuation formula (6), which in the setting of ridge regression is exact. Our goal is not to re-derive known high-dimensional risk formulas, but to highlight their two-operator structure: test error decomposes into (i) train/test spectral scales and (ii) an explicit eigenspace-alignment kernel. This makes covariate shift and multiple descent analyzable as overlap phenomena. Letting the label noise supply the perturbation, the injected noise $z := d\nabla_w \mathcal{L}_{\text{train}}(w_0, \xi)$ and displacement covariance $C_{\text{train}} := \mathbb{E}\left[(H_{\text{train}}^{-1} z)(H_{\text{train}}^{-1} z)^\top\right]$ are

$$z = -\tfrac{\sqrt{d}}{m} X^\top \xi, \qquad C_{\text{train}} = \sigma^2 \alpha^{-1} \hat{\Sigma}_{\text{train}}(\hat{\Sigma}_{\text{train}} + \lambda I)^{-2}, \tag{10}$$

where $\alpha := m/d$ is the sampling ratio. The test loss increment $\Delta\mathcal{L}$ is obtained by substituting into the overlap formula (6). The training-side operators $\hat{\Sigma}_{\text{train}}$, $H_{\text{train}} = \hat{\Sigma}_{\text{train}} + \lambda I$ and $C_{\text{train}}$ commute and share eigenvectors, so for simplicity all formulas are written in terms of $\hat{\Sigma}_{\text{train}}$:

$$\mathbb{E}\left[\Delta\mathcal{L}\right] = \tfrac{\sigma^2}{2\alpha} \iint \lambda_1 \frac{\lambda_2}{(\lambda_2 + \lambda)^2} O_{\Sigma_{\text{test}}, \hat{\Sigma}_{\text{train}}}(\lambda_1, \lambda_2) \, \mu_{\Sigma_{\text{test}}}(d\lambda_1) \, \mu_{\hat{\Sigma}_{\text{train}}}(d\lambda_2), \tag{11}$$

where $\mu_{\Sigma_{\text{test}}}$ and $\mu_{\hat{\Sigma}_{\text{train}}}$ are the empirical spectral measures, and $O_{\Sigma_{\text{test}}, \hat{\Sigma}_{\text{train}}}(\lambda_1, \lambda_2)$ is the eigenvector-overlap function (see (18)). Since we will be interested primarily in the ridgeless limit $\lambda \to 0$, we will loosely refer to $\hat{\Sigma}_{\text{train}}$ as the train Hessian. See Appendix C for detailed derivations.

The fundamental conclusion from (11) that we will apply toward analyzing covariate shift and multiple descent is that error is large when training perturbations induce large variance (small training eigenvalue $\lambda_2$) in directions that align strongly (large $O(\lambda_1, \lambda_2)$) with directions of large test loss sensitivity (large test eigenvalue $\lambda_1$).

In Appendix C, using techniques from operator-valued free probability we derive asymptotically exact expressions for $\mathcal{L}_{\text{test}}, \Delta\mathcal{L}$ and the overlap function $O_{H_{\text{train}}, H_{\text{test}}}$ in proportional asymptotics where $m, d \to \infty$ with $\alpha := m/d$ fixed. The main conceptual contribution of this work is that while the spectral densities of train/test operators set the relevant scales, it is their relative orientation—as quantified by the overlap function—that determines how displacements translate into test loss. We illustrate these points in two settings: first, a simple covariate shift experiment that provides geometric intuition and positions $O_{\Sigma_{\text{test}}, \Sigma_{\text{train}}}$ as the natural object quantifying shift; second, the puzzle of multiple descent (cf. Mel & Ganguli (2021)), where the overlap function allows a full analytical account. For clarity, in both settings we use the simplest possible model of anisotropic data: the "two-scale" covariance with spectral measure

$$\mu_\Sigma := p_1 \delta_{s_1^2} + p_2 \delta_{s_2^2}. \tag{12}$$

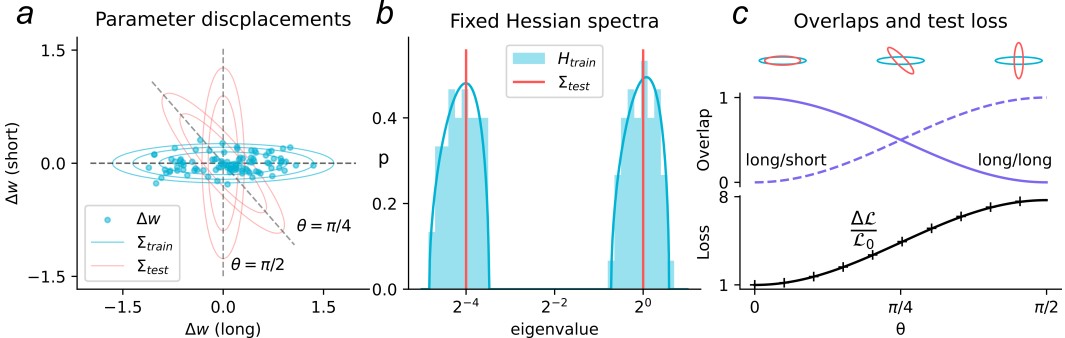

Figure 1: Isospectral shifts isolate overlap effects in covariate shift. (a) Two-dimensional slice of parameter space. Points: individual parameter displacements due to label noise. Lines show train (cyan) and test (red) Hessians with precisely controlled alignment $\theta$. (b) Eigenspaces are rotated while Hessian spectra are fixed. Blue line shows $d, m \to \infty$ theory. (c) For small $\theta$, large displacements in learned parameters are aligned with the low-eigenvalue directions of the test Hessian (aligned ellipses at top left, and purple overlap lines), and error is small (black line). For large $\theta$, large displacement directions are aligned with sensitive directions of the test loss and test error is large (black line). Lines: theory; crosses: average from simulations. $d, \alpha, \lambda, \sigma = 10^2, 10, 10^{-4}, 10^{-1/2}$.

### 3.2.1 COVARIATE SHIFT INCREASES LOSS THROUGH TRAIN-TEST MISALIGNMENT

Equation (11) expresses $\Delta\mathcal{L}$ in terms of the overlap function $O_{\Sigma_{\text{test}}, \hat{\Sigma}_{\text{train}}}$. Relative to the population overlap $O_{\Sigma_{\text{test}}, \Sigma_{\text{train}}}$, this overlap is deformed by the finite sampling ratio of the training set (cf. transfer law of Theorem 2). In Appendix C.3 we use the transfer law to state an explicit formula for $O_{\Sigma_{\text{test}}, \hat{\Sigma}_{\text{train}}}$, and then prove the following:

**Theorem 3.** *As $m, d \to \infty$ with $\alpha$ fixed, the asymptotic test loss increment satisfies*

$$\mathbb{E}[\Delta\mathcal{L}] \to \frac{\sigma^2}{2\alpha} \frac{d\tilde{\lambda}}{d\lambda} \iint \lambda_1 \frac{\lambda_2}{(\lambda_2 + \tilde{\lambda})^2} O_{\Sigma_{\text{test}}, \Sigma_{\text{train}}} (\lambda_1, \lambda_2) \, \mu_{\Sigma_{\text{test}}} (d\lambda_1) \, \mu_{\Sigma_{\text{train}}} (d\lambda_2), \tag{13}$$

*where $\tilde{\lambda}$ is the effective regularization defined by the self-consistent equation:*

$$\tilde{\lambda} := \frac{\lambda}{r(-\lambda)}, \qquad r(z) = \left(1 - \frac{1}{\alpha} \int \frac{t}{z - t\, r(z)} d\mu_{\Sigma_{\text{train}}} (t)\right)^{-1}. \tag{14}$$

Equation (13) parallels (11) but averages out all training randomness to express $\Delta\mathcal{L}$ purely in terms of the population operators $\Sigma_{\text{train}}, \Sigma_{\text{test}}$. Most importantly, (13) illustrates how $O_{\Sigma_{\text{test}}, \Sigma_{\text{train}}}$—as the only quantity that can change under isospectral transformations to $\Sigma_{\text{train}}, \Sigma_{\text{test}}$—captures bona fide two-loss geometric effects that are invisible from either loss geometry in isolation.

To illustrate this point, we perform a simple experiment where both $\Sigma_{\text{train}}, \Sigma_{\text{test}}$ have fixed two-level spectra (12) with scales $s_1^2, s_2^2 = 2^0, 2^{-4}$ and equal multiplicities. $\lambda = 10^{-4}$ and $\alpha = m/d = 10$ so that $H_{\text{train}} \approx \hat{\Sigma}_{\text{train}} \approx \Sigma_{\text{train}}$, while $H_{\text{test}} = \Sigma_{\text{test}}$. Fig. 1(a) shows the distribution of learned parameter displacements for different label noise realizations. As predicted, displacements have larger variance along long directions of $C_{\text{train}} \approx \sigma^2 \Sigma_{\text{train}}^{-1}/\alpha$, corresponding to low-curvature directions of the train Hessian. At the same time, the test loss contours are determined by the test Hessian $\Sigma_{\text{test}}$. We construct a controlled perturbation in which $\Sigma_{\text{test}}$ is systematically rotated with respect to $\Sigma_{\text{train}}$ while all spectra are kept fixed (b), isolating the effect of overlaps. Fig. 1(c) demonstrates the consequence of varying overlap. When the long directions of $\Sigma_{\text{train}}$ align with the long directions of $\Sigma_{\text{test}}$ ($\theta = 0$), displacements occur in directions where the test error is relatively flat, yielding low excess test loss (Fig. 1(c), left column). In contrast, when the same train-long directions align with test-short directions ($\theta = \pi/2$), the same magnitude of parameter displacement is heavily penalized, and the test loss rises sharply (Fig. 1(c), right column). This simple experiment illustrates the central role of eigenvector overlaps in the context of covariate shift.

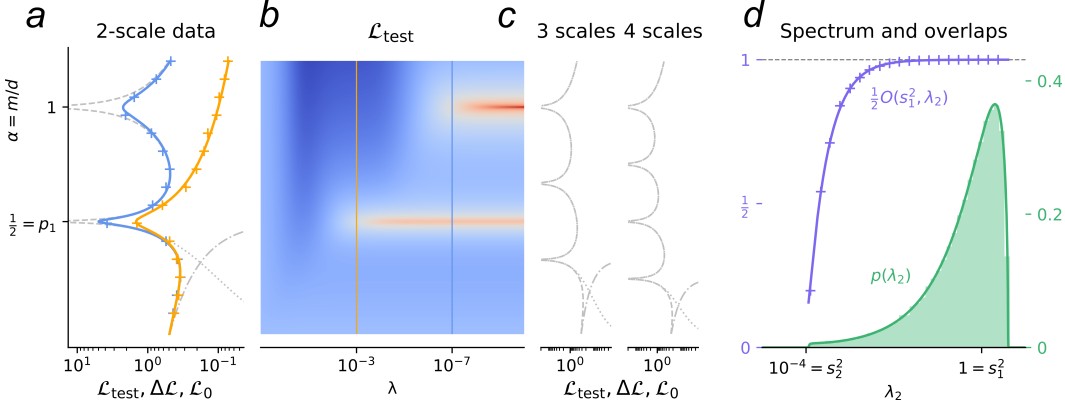

Figure 2: Multiple descent in ridge regression. (a) Loss as a function of the measurement density $\alpha = m/d$ for two-scale data. Note the peaks at critical values of $\alpha = 1/2, 1$. Solid lines: theory; crosses: simulations with $d = 5000$. Dashed, dotted, and dash-dotted lines show theory $\mathcal{L}_{\text{test}}, \Delta\mathcal{L}, \mathcal{L}_0$ in the limit that the lower scale $s_2 \to 0$, where bumps become true singularities. (b) Theory $\mathcal{L}_{\text{test}}(\alpha, \lambda)$. Gold and blue lines indicate slices shown in panel (a). (c) 3 and 4 scale data which exhibit 3 and 4 peaks; legend same as (a). (d) Green histogram: empirical spectral density of the train Hessian $X^\top X/m$ at $\alpha = 0.496$; solid green line: theory. Purple line: overlap function, $O(s_1^2, \lambda_2)/2$, giving overlap between a train eigenspace at eigenvalue $\lambda_2$ with the entire large-eigenvalue test space (ie. $s_1^2$). Note strong overlap for high train/test eigenspaces.

### 3.2.2 MULTIPLE DESCENT IS EXPLAINED BY TRAIN-TEST OVERLAPS

Double descent is a well-established phenomenon in machine learning in which test error exhibits a non-monotonic dependence on model size. More recently, several authors have described an extension of this effect, termed multiple descent, which arises in settings where input data are highly anisotropic and the covariance spectrum contains multiple separated scales (see introduction). Fig. 2(a,b) illustrate multiple descent for two-scale data with $s_1, s_2 = 1, 10^{-2}$, while panel c shows how a larger number of separated scales can create additional peaks in test error (see caption for details).

For a two-level covariance, the overlap function is determined by the solution to a cubic polynomial that is easily solved numerically (Appendix C). Fig. 2(d) shows the spectrum of the train Hessian (green histogram and theory line), and the overlap function (18), indicating overlap of a training eigenspace at eigenvalue $\lambda_2$ with the large-eigenvalue ($s_1^2$) eigenspace of the test Hessian. Theoretical and empirical overlaps are in excellent agreement (purple line and crosses).

The peaks of multiple descent are easily understood in terms of eigenvector overlaps. Fig. 3 reports the error, training spectrum, and overlap map for the two-scale covariance model of Fig. 2. The test-loss curve shows two singularities at critical sampling densities $\alpha = m/d$ (a). At the same densities the training spectrum undergoes phase transitions: at $\alpha = 1/2$ an initially unimodal density splits into two bands centered near $s_1^2$ and $s_2^2$, and at $\alpha = 1$ the lower $s_2^2$ band develops a near-zero component (Fig. 3(b)). The corresponding overlap map $O(\lambda_1, \lambda_2)$ is approximately block-diagonal: modes near $s_1^2$ align predominantly with the $s_1^2$ test subspace, and modes near $s_2^2$ with the $s_2^2$ subspace (Fig. 3(c)). Thus, the first error spike occurs when near-null training directions overlap the sharp test subspace, whereas the second arises when an even smaller training component overlaps the flat subspace but with variance large enough to dominate its small curvature. Fig. 3(d) provides a geometric schematic of the alignment of top and bottom eigenspaces of $H_{\text{train}}, H_{\text{test}}$ throughout this sequence. Until line 5, the minimum eigenvalue of $H_{\text{train}}$ always decreases as a function of $\alpha$—which, according to a spectrum-only analysis, should increase test error. Yet the error actually *decreases* between horizontal lines 3 and 4, precisely because the lowest train eigenspaces begin to overlap predominately with the low test eigenspace.

Summarizing, multiple descent arises from the interplay of (i) training components developing near-zero eigenvalues as $\alpha$ varies, and (ii) which test directions these overlap with—sharp or flat, illustrating the potentially extreme impact of (mis-)aligned train and test loss geometry.

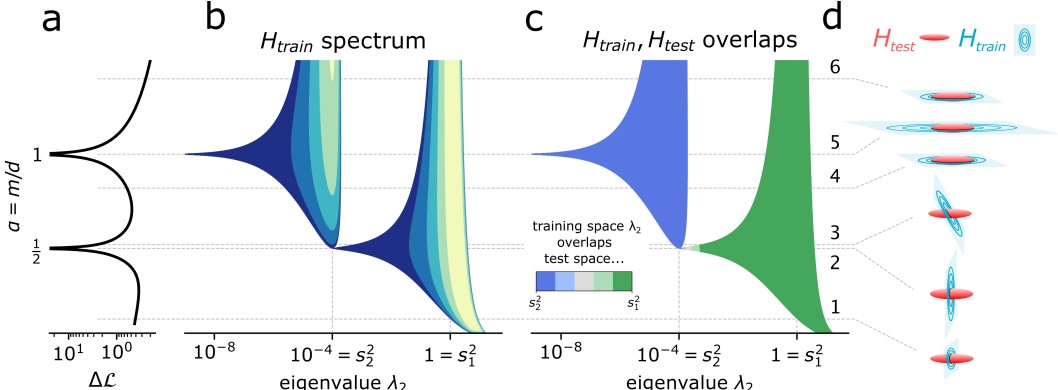

Figure 3: Multiple descent is explained by train-test Hessian overlaps. (a) $\Delta\mathcal{L}$ due to label noise ($\lambda, s_2^2 \to 0$ limits taken to illustrate true singularities; see Appendix C.5.1 for details). (b) Spectral density of $H_{\text{train}}$ as a function of $\alpha$ (each density normalized to a maximum of 1 and quantized). For high sampling density $\alpha$, the training density acquires two components roughly centered on the true underlying data scales $s_1^2, s_2^2$. (c) The overlap function of $H_{\text{train}}, H_{\text{test}}$ as a function of $\alpha$. For small $\alpha$, all nonzero $H_{\text{train}}$ spaces overlap strongly with the large eigenspace of $H_{\text{test}}$. As $\alpha$ approaches the first critical value, a new spectral component appears, whose eigenspaces overlap almost entirely with the small eigenspace of $H_{\text{test}}$. (d) Minimal model of train-test Hessian geometry. Cyan plane and ellipses represent the top and bottom eigenspaces of $H_{\text{train}}$. Red ellipsoid represents level sets of test error. Error is controlled by both train variance magnitude and overlap onto test spaces.

### 3.3 LOCAL THEORY PREDICTS MLP GENERALIZATION AND LEARNING DYNAMICS

To test the quantitative predictions of the quadratic two-loss theory in a controlled nonconvex setting, we trained small, constant width multilayer perceptrons (MLPs) to reproduce the responses of an MLP teacher network. Student networks were batch trained for a large number of iterations to ensure near convergence to the noiseless training loss minimum. Noise was then added to the training set and the network was trained further—beginning from the initial trained state to determine the effect of the noise on the initial local minimum. After training, the training loss increment was computed and compared to prediction of the local quadratic theory. Fig. 4(a,b) show the measured test loss increment against the local quadratic prediction for several orders of magnitude of input (a) and label (b) noise strength. All later panels refer to the noise setting corresponding to the red point in (b).

Fig. 4(c) illustrates inverse Hessian filtering due to training dynamics. The gradient noise induced by the label noise has covariance $\mathbb{E}[zz^\top]$. The purple scatter represents the overlap function of $\mathbb{E}[zz^\top]$ and $H_{\text{train}}$. Dot $x, y$ position is given by $H_{\text{train}}, \mathbb{E}[zz^\top]$ eigenvalue and size is proportional to overlap. Note that the gradient noise and train Hessian are strongly aligned. After training, the parameter displacement covariance predicted by quadratic approximation is $C_{\text{train}} := \mathbb{E}\left[(H_{\text{train}}^{-1}z)(H_{\text{train}}^{-1}z)^\top\right]$. The overlap function of $H_{\text{train}}$ and the actual post-training covariance is plotted in red. Note how in accordance with quadratic predictions, variance is strongly inflated/attenuated along low/high eigendirections of $H_{\text{train}}$—a phenomenon we refer to as inverse Hessian filtering. The large displacements do not translate into large test error since the train and test Hessians are well aligned (Fig. 6), meaning displacements occur primarily along low test Hessian (loss-insensitive) directions.

To provide geometric intuition, loss landscape slices are shown in Fig. 4(d) for $\mathcal{L}_{\text{train}}(w, 0)$, $\mathcal{L}_{\text{train}}(w, \epsilon)$, and $\mathcal{L}_{\text{test}}$. A single 2D slice was chosen to contain the unperturbed minimum $w_0$ (white crosses), perturbed minimum (white stars), and parameters predicted by the local quadratic approximation (white "Y"s). Local geometry also predicts local gradient descent dynamics well (Appendix E.1; Fig. 7). Together, these results validate the predictions of the two-loss local theory.

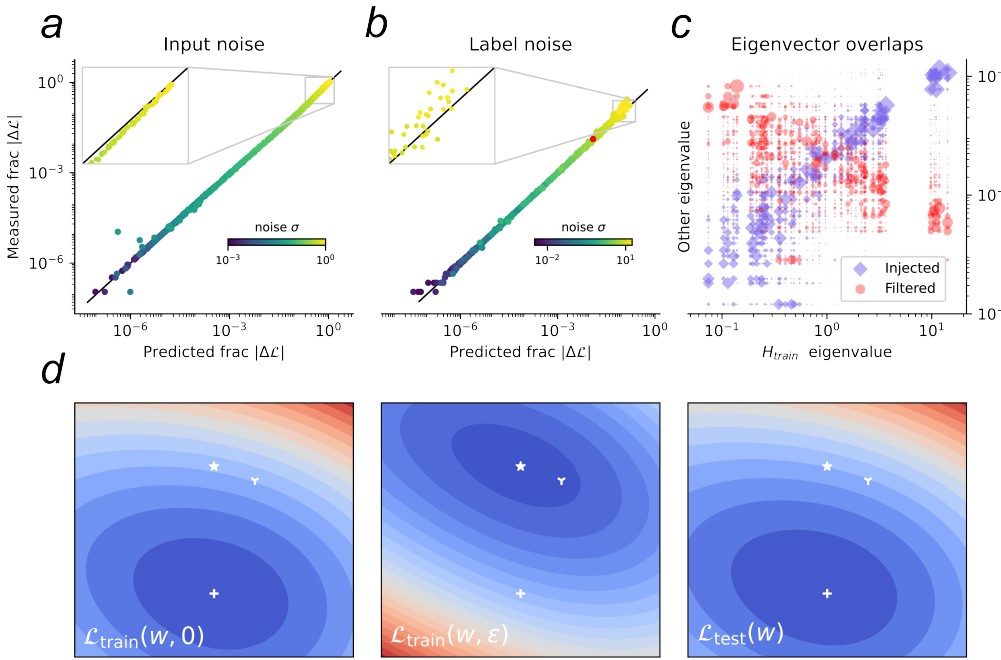

Figure 4: Validation of local fluctuation law in MLPs. Layer widths for both student and teacher were (5,5,5,1); nonlinearity: tanh; teacher network has gaussian weights with scale $4/\sqrt{d_{in}}$ for each layer; loss: MSE with $\ell_2$ parameter $\lambda = 1$. (a,b) Predicted vs measured perturbation-induced $\Delta\mathcal{L}/\mathcal{L}_0$ for increasing input (a) or label (b) noise amplitude $\sigma$. (c) Eigenvector overlap function between $H_{\text{train}}$ and the injected noise $\mathbb{E}[zz^\top]$ (purple), and post-learning filtered noise $C_{\text{train}}$ (red). Note how learning inflates/dampens variance along the low/high $H_{\text{test}}$ eigenspaces. (d) 2-dimensional loss landscapes for 1 example simulation: noiseless $\mathcal{L}_{\text{train}}$ (left), perturbed $\mathcal{L}_{\text{train}}$ (middle), and $\mathcal{L}_{\text{test}}$ (right). Cross: noiseless training minimum; star: minimum of perturbed $\mathcal{L}_{\text{train}}$ (ie. the new learned minimum); tri-star: parameters predicted by quadratic theory.

## 3.4 CALCULATION OF OVERLAP FUNCTIONS FOR LARGE SCALE NETWORKS

Applying our theory to modern networks requires estimating the overlap function between the training and test operators. These operators have dimension equal to the number of parameters—often millions to billions—so any approach that forms them explicitly is infeasible.

Here we give a brief overview of our approach, deferring details to Appendix F. We apply two separate algorithms, one for computing overlaps among outlier eigenspaces and another for the remaining "bulk" spaces. Outlier eigenvectors are straightforward to obtain using subspace iteration (Appendix F.2; cf. Papyan (2019)); overlaps can then be computed directly. For the bulk eigenspaces, we generalize a well-known approach to spectral density estimation known as the kernel polynomial method (KPM; Algorithm 1 in Appendix F.3).

Given self-adjoint matrices $A, B \in \mathbb{R}^{d \times d}$ and arbitrary smoothing kernels $G(x; \sigma)$ of width $\sigma$, the smoothed total eigenvector overlap of $A, B$ at eigenvalues $\lambda_1, \lambda_2$ can be written

$$\bar{\text{tr}}\,[G_{A,\lambda_1} G_{B,\lambda_2}] = \frac{1}{d^2} \sum_{i,j=1}^{d} G(\lambda_{A,i} - \lambda_1; \sigma)\, G(\lambda_{B,j} - \lambda_2; \sigma) \left[ d\,(v_{A,i} \cdot v_{B,j})^2 \right], \quad (15)$$

where $G_{A,\lambda_1} := G(A - \lambda_1 I; \sigma)$ and similarly for $G_{B,\lambda_2}$. To obtain the normalized overlap function treated above, one simply divides by the ($G$-smoothed) spectral densities of $A, B$ at $\lambda_1, \lambda_2$.

Computing the trace in (15) is prohibitively expensive for large $A, B$, and so we resort to Hutchinson trace estimation, which approximates $\text{tr}\,[X]$ with the average of $v^\top X v$ for several samples of $v \sim \mathcal{N}(0, I)$. To ensure the trace is positive, instead of approximating (15), we use

$$\bar{\text{tr}}\,[G_{A,\lambda_1} G_{B,\lambda_2}] = \bar{\text{tr}}\,[G_{A,\lambda_1}^{1/2} G_{B,\lambda_2} G_{A,\lambda_1}^{1/2}] = \mathbb{E}_v \|G_{B,\lambda_2}^{1/2} G_{A,\lambda_1}^{1/2} v\|^2. \quad (16)$$

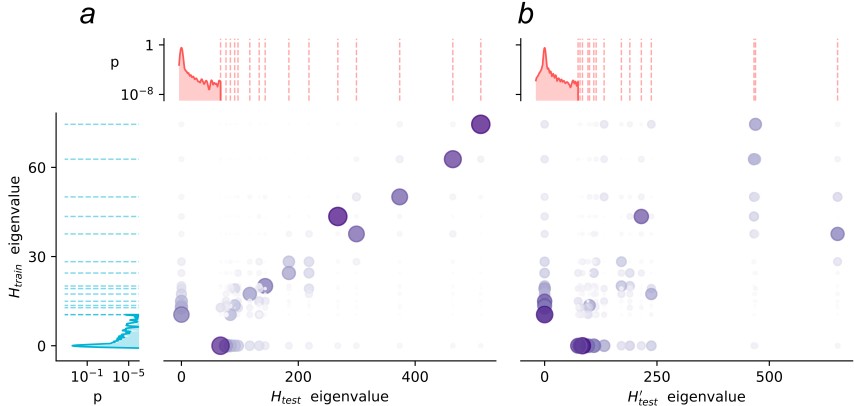

Figure 5: Overlap function for $H_{\text{train}}, H_{\text{test}}$, and class imbalanced Hessian $H'_{\text{test}}$ for ResNet-20 trained on CIFAR-10. Cyan/red data: spectra of $H_{\text{train}}, H_{\text{test}}$. Dashed lines indicate outlier eigenvalues. Purple scatters show overlap between each pair of eigenspaces/bulk spaces. Size and color reflect overlap magnitude. (a) $H_{\text{train}}, H_{\text{test}}$ overlaps. Note strong alignment indicated by large overlaps along the diagonal. (b) $H_{\text{train}}, H'_{\text{test}}$ overlaps. A large fraction of each Hessian's outlier energy is lost in low-outlier and bulk spaces of the other, indicating poor alignment.

The KPM proceeds by taking the smoothing kernel $G(x; \sigma)$ to be gaussian of width $\sigma$, and then approximates $G_{A,\lambda_1}^{1/2}, G_{B,\lambda_2}^{1/2}$ using truncated Chebyshev series. (Kernel width and approximation degree $K$ are chosen so that the truncated series sufficiently dampens the large-multiplicity near-0 eigenspaces; see Appendix F.3.) Thus (16) can be evaluated in terms of the vectors $T_i(B) T_j(A) v$, where $T_k$ is the $k^{th}$ Chebyshev polynomial. These vectors in turn can be generated efficiently via Chebyshev recurrences using only matrix-vector products (see Appendix F for detailed treatment and application to synthetic data).

We ran a simple controlled experiment to demonstrate the scalability of our Hessian-overlap algorithms on a modern network and to illustrate how a common form of domain shift—class imbalance in the test set—produces a clear change in two-loss geometry. We used a publicly available CIFAR-10 pretrained ResNet-20 checkpoint from Chen (top-1 test accuracy: $92.6\%$). The train Hessian was estimated from 5000 examples and fixed throughout the experiment. Two-loss geometry was then compared between two scenarios: a class-balanced test Hessian estimated from 5000 randomly selected test images, and a class-imbalanced Hessian from images with class labels 0, 1 and 2. Spectra, estimated using subspace iteration and the Lanczos algorithm, are shown in Fig. 5(a) (train in cyan; test in red). Non-outlier eigenspaces were grouped into a single bulk space for clarity. The strong alignment observed between the train and balanced test Hessians largely disappears when the test set is made unbalanced (purple scatters; bulk overlaps, omitted for space, exhibit similar pattern; Fig. 10). All Hessian-vector products were computed using standard PyTorch autograd on commodity hardware, with total runtime of a few hours. Runtimes are essentially linear in the model size and number of examples, underscoring the scalability of our method (see complexity analysis in F.3).

## 4 DISCUSSION

We show how, within a two-loss geometric framework, overlaps occupy a central role linking optimization geometry, random matrix theory, and practical machine learning phenomena. We derive novel theoretical tools for computing overlaps, illustrate through several examples how spectra set curvatures, while eigenvector overlaps route variance into error—unifying covariate shift and multiple descent—and develop scalable estimators for overlap analysis in large models. A natural application of two-loss geometry is as a diagnostic tool for explaining why some domain shifts are more harmful than others. Promising future directions include tracking Hessian overlaps through training time, and *alignment-aware optimization* that attempts to improve generalization by encouraging strong eigenvector alignment between, eg., train and validation Hessians.

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

## A  STATEMENT ON LARGE LANGUAGE MODEL USE

Large language models were used to polish writing and in conjunction with other tools to discover relevant published work.

## B  THEORETICAL FOUNDATIONS

### B.1  EIGENVECTOR OVERLAP FUNCTION

To connect the finite-$d$ decomposition to random-matrix and free-probability tools, we now express eigenvector overlaps in a kernelized trace form amenable to free-probabilistic methods. Let $X, Y$ be symmetric $d \times d$ matrices with eigendecompositions

$$X = \sum_{i=1}^{d} \lambda_i^X \, u_i u_i^\top, \qquad Y = \sum_{j=1}^{d} \lambda_j^Y \, v_j v_j^\top.$$

For bounded functions $f, g$,

$$\bar{\mathrm{tr}}\big[f(X)\,g(Y)\big] = \frac{1}{d^2} \sum_{i=1}^{d} \sum_{j=1}^{d} f(\lambda_i^X)\, g(\lambda_j^Y)\, \big[d\,(u_i^\top v_j)^2\big]. \tag{17}$$

If $f$ and $g$ are sharply peaked around $\lambda_1$ and $\lambda_2$, the sum concentrates on overlaps between eigenvectors with eigenvalues near $(\lambda_1, \lambda_2)$.

A convenient choice is the Poisson kernel

$$K(x; \mu, \sigma) := \frac{1}{\pi} \frac{\sigma}{(x - \mu)^2 + \sigma^2},$$

with center $\mu$ and width $\sigma > 0$. We define the overlap function

$$O(\lambda_1, \lambda_2) := \lim_{\sigma_1, \sigma_2 \to 0} \frac{\bar{\mathrm{tr}}\big[K(X; \lambda_1, \sigma_1)\, K(Y; \lambda_2, \sigma_2)\big]}{\bar{\mathrm{tr}}\big[K(X; \lambda_1, \sigma_1)\big]\, \bar{\mathrm{tr}}\big[K(Y; \lambda_2, \sigma_2)\big]}. \tag{18}$$

The denominator normalizes the total weight in the sum (17) to one, so $O(\lambda_1, \lambda_2)$ is the weighted average of the (scaled) squared overlaps $d\,(u_i^\top v_j)^2$ over eigenpairs near $(\lambda_1, \lambda_2)$. Now (17) can be rewritten

$$\bar{\mathrm{tr}}\,[f(X)\,g(Y)] = \iint f(\lambda_1)\, g(\lambda_2)\, O(\lambda_1, \lambda_2)\, d\mu_X(\lambda_1)\, d\mu_Y(\lambda_2). \tag{19}$$

In fact, another way to define the overlap function is to write $\mu_{X,Y}$ for the measure taking $f, g \mapsto \bar{\mathrm{tr}}\,[f(X)\,g(Y)]$ and then defining $O(\lambda_1, \lambda_2)$ to be the function making (19) hold, ie. $O = \frac{d\mu_{X,Y}}{d\mu_X \otimes \mu_Y}$.

## B.2 OVERLAP FLUCTUATION LAW

Here we prove the fluctuation law (6). Equation (5), which uses the quadratic surrogate losses $\mathcal{L}_{\text{train}}^{\text{quad}}, \mathcal{L}_{\text{test}}^{\text{quad}}$ to compute the test loss increment, reads

$$\Delta \mathcal{L} = -\tfrac{1}{d} z_{\text{test}}^{\top} H_{\text{train}}^{-1} z + \tfrac{1}{2d} z^{\top} H_{\text{train}}^{-1} H_{\text{test}} H_{\text{train}}^{-1} z. \tag{20}$$

Noting that $\Delta w = -H_{\text{train}}^{-1} z$, under the assumption that $\mathbb{E}[\Delta w] = 0$, one clearly has

$$\mathbb{E}[\Delta \mathcal{L}] = \tfrac{1}{2} \mathbb{E} \, \bar{\text{tr}}[H_{\text{test}} H_{\text{train}}^{-1} z z^{\top} H_{\text{train}}^{-1}] = \tfrac{1}{2} \bar{\text{tr}}[H_{\text{test}} C_{\text{train}}]. \tag{21}$$

All that's left is to show that the last trace can be expressed in the integral form (6). Letting $(\lambda_i^{\text{test}}, u_i^{\text{test}}), (\lambda_j^{\text{train}}, u_j^{\text{train}})$ be the eigenvalues/eigenvectors of $H_{\text{test}}, C_{\text{train}}$,

$$\tfrac{1}{2d} \text{tr}[H_{\text{test}} C_{\text{train}}] = \tfrac{1}{2} \tfrac{1}{d^2} \sum_{i=1}^{d} \sum_{j=1}^{d} \lambda_i^{\text{test}} \lambda_j^{\text{train}} [d \, (u_i^{\text{test}} \cdot u_j^{\text{train}})^2]. \tag{22}$$

Defining the overlap measure

$$\nu := \frac{1}{d^2} \sum_{i=1}^{d} \sum_{j=1}^{d} [d \, (u_i^{\text{test}} \cdot u_j^{\text{train}})^2] \, \delta_{(\lambda_i^{\text{test}}, \lambda_j^{\text{train}})}, \tag{23}$$

equation (22) can be written

$$\tfrac{1}{2d} \text{tr}[H_{\text{test}} C_{\text{train}}] = \tfrac{1}{2} \iint \lambda_1 \lambda_2 \nu \, (d\lambda_1, d\lambda_2). \tag{24}$$

$\nu$ is absolutely continuous with respect to $\mu_{\text{test}} \otimes \mu_{\text{train}}$, and so we may define the Radon-Nikodym derivative $O(\lambda_1, \lambda_2) = \frac{d\nu}{d\mu_{\text{test}} \otimes \mu_{\text{train}}}(\lambda_1, \lambda_2)$ so that

$$\tfrac{1}{2d} \text{tr}[H_{\text{test}} C_{\text{train}}] = \tfrac{1}{2} \iint \lambda_1 \lambda_2 O(\lambda_1, \lambda_2) \mu_{\text{test}}(d\lambda_1) \mu_{\text{train}}(d\lambda_2). \tag{25}$$

On any atom $(\lambda_i^{\text{test}}, \lambda_j^{\text{train}})$,

$$O(\lambda_i^{\text{test}}, \lambda_j^{\text{train}}) = \frac{\nu \left( \{(\lambda_i^{\text{test}}, \lambda_j^{\text{train}})\} \right)}{\mu_{\text{test}}(\{\lambda_i^{\text{test}}\}) \mu_{\text{train}}(\{\lambda_j^{\text{train}}\})} = d \, (u_i^{\text{test}} \cdot u_j^{\text{train}})^2, \tag{26}$$

as desired.

### B.2.1 SURROGATE-FREE FORMULATION

For completeness, we derive the fluctuation law without the use of quadratic surrogate losses by making a minor modification to the train Hessian. As before, let $\mathcal{L}_{\text{train}}(w, \epsilon)$ and $\mathcal{L}_{\text{test}}(w)$ denote the train and test losses, assumed twice differentiable in $w$, and let $w_0$ be the unperturbed minimizer of $\mathcal{L}_{\text{train}}(w, 0)$. For small perturbation $\epsilon$, write $\Delta w = w(\epsilon) - w_0$ for the exact displacement. By the fundamental theorem of calculus along the line segment $w_0 + t\Delta w$,

$$\nabla_w \mathcal{L}_{\text{train}}(w_0 + \Delta w, \epsilon) = \nabla_w \mathcal{L}_{\text{train}}(w_0, \epsilon) + \left[ \int_0^1 \nabla_w^2 \mathcal{L}_{\text{train}}(w_0 + t\Delta w, \epsilon) \, dt \right] \Delta w.$$

Define the *effective train Hessian*

$$H_{\text{train}}^{\text{eff}} := \int_0^1 \nabla_w^2 \mathcal{L}_{\text{train}}(w_0 + t\Delta w, \epsilon) \, dt.$$

The perturbed optimality condition $\nabla_w \mathcal{L}_{\text{train}}(w_0 + \Delta w; \epsilon) = 0$ therefore yields the exact displacement equation

$$\Delta w = -(H_{\text{train}}^{\text{eff}})^{-1} z,$$

where $z = \nabla_w \mathcal{L}_{\text{train}}(w_0, \epsilon)$. Thus, $\Delta w$ is obtained by the same inverse-Hessian filtering law as in the quadratic case, with $H_{\text{train}}$ replaced by $H_{\text{train}}^{\text{eff}}$.

For the test-loss increment, apply an ordinary Taylor expansion at $w_0$:

$$\mathcal{L}_{\text{test}}(w_0 + \Delta w) = \mathcal{L}_{\text{test}}(w_0) + \tfrac{1}{d} z_{\text{test}} \cdot \Delta w + \tfrac{1}{2d} \Delta w^\top H_{\text{test}} \Delta w + O(\|\Delta w\|^3),$$

where $z_{\text{test}} = d \nabla_w \mathcal{L}_{\text{test}}(w_0)$ and $H_{\text{test}} = d \nabla_w^2 \mathcal{L}_{\text{test}}(w_0)$ are evaluated at the unperturbed point and are independent of $\epsilon$. Substituting the displacement equation, as in the surrogate case one obtains:

$$\Delta \mathcal{L} = -\tfrac{1}{d} z_{\text{test}}^\top (H_{\text{train}}^{\text{eff}})^{-1} z + \tfrac{1}{2d} z^\top (H_{\text{train}}^{\text{eff}})^{-1} H_{\text{test}} (H_{\text{train}}^{\text{eff}})^{-1} z + O(\|\epsilon\|^3).$$

Taking expectations over the perturbation, the quadratic term has the same form as in Theorem 1, $\tfrac{1}{2} \bar{\text{tr}}[H_{\text{test}} C_{\text{train}}]$, where $C_{\text{train}} = \mathbb{E}\left[(H_{\text{train}}^{\text{eff}})^{-1} zz^\top (H_{\text{train}}^{\text{eff}})^{-1}\right]$, and so after replacing $H_{\text{train}}$ with $H_{\text{train}}^{\text{eff}}$, one obtains a formally identical overlap decomposition as in (6) in terms of test-(effective) train eigenvector overlaps.

### B.2.2 Noisy gradient descent

Consider gradient descent with small additive noise at each iteration, and take its continuum limit. In the local quadratic regime, such noisy gradient descent can be modeled by the linear stochastic differential equation (SDE)

$$dw_t = -H_{\text{train}} w_t dt + \Sigma^{1/2} dB_t,$$

where $\Sigma$ is the noise covariance and $B_t$ is standard Brownian motion. Its stationary covariance $C_{\text{train}}$ satisfies the Lyapunov equation

$$H_{\text{train}} C_{\text{train}} + C_{\text{train}} H_{\text{train}} = \Sigma.$$

This covariance is curvature-filtered—variance is suppressed along high-curvature directions and amplified along shallow ones—similarly to the perturbation-induced $C_{\text{train}}$ used in the main text. Substituting this covariance into the trace formula yields the same overlap fluctuation law and the same two-loss spectral-overlap decomposition, with no modifications to the framework required.

### B.3 Free transfer law

Here we prove the following free transfer law for overlap functions:

**Theorem** (Theorem 2, Free transfer law). *Let $\hat{B}$ combine $B$ with a source of noise $X$ that is free from $A, B$. Then*

$$O_{A,\hat{B}}(a, \hat{b}) = \int O_{A,B}(a, b) O_{B,\hat{B}}(b, \hat{b}) \, \mu_B \, (db). \tag{27}$$

Note that despite its simple and appealing form, this relationship does not hold for general triples of matrices $A, B, C$—it suffices to check on finite dimensional matrices with simple spectra. While the coefficients of $C$'s eigenvectors in the eigenbases of $A, B$ do follow a change of basis law resembling formula (27), recall that the overlap function encodes the squared coefficients rather than the raw coefficients themselves. This relationship holds specifically because of the freeness relationship we've assumed.

To prove (27), start by noting that all of the spectral and overlap information for two matrices $A, B$ is contained in the measure on $\mathbb{R}^2$ corresponding to the functional

$$\mu_{A,B} : f, g \mapsto \tau\left[f(A) g(B)\right].$$

For example, setting $g \to 1$ and $f(x) = x^m$ gives access to all moments of $A$, and therefore to its spectrum, and similarly for $B$, while the overlap function is precisely $O_{A,B} = \frac{d\mu_{A,B}}{d\mu_A \otimes \mu_B}$, the Radon-Nikodym derivative of the joint measure with respect to the marginals.

For the remainder of this section, we will work in an abstract free probability space rather than with concrete matrices. Let $(\mathcal{M}, \tau)$ be a $W^*$-probability space (in our application this corresponds to the space of $d \times d$ matrices with $\tau = \mathbb{E}\,\bar{\text{tr}}$). See Mingo & Speicher (2017) for details.

Let $A, B, \hat{B}, X \in \mathcal{M}$ be random variables and consider the problem of determining the overlap function $O_{A,\hat{B}}$, where $\hat{B} := F(B, X)$, where $X$ is a source of noise that is free from $A, B$. As noted

above, all of the overlap information for the three possible pairs of variables $A, B, \hat{B}$ is contained in the joint measures

$$\mu_{A,\hat{B}}, \mu_{A,B}, \mu_{B,\hat{B}},$$

supported on (some subset of) $\mathbb{R}^2$. We are free now to treat these measures as ordinary probability measures of two scalar random variables. Denote by $\langle \cdot \rangle$ these scalar expectations. We now make use of the following two identities:

$$\langle f(x) g(y) \rangle_{(x,y) \sim \mu_{X,Y}} = \tau \left[ f(X) g(Y) \right], \tag{28}$$

and that for each $g$ in a suitably broad class of functions (eg. at minimum all Poisson kernels), there is another function $L_g$ depending linearly on $g$ encoding the "expectation of $g\left(\hat{B}\right) = g\left(F(B, X)\right)$ over $X$, conditioned on $B$", ie. such that

$$\tau \left[ f(A, B) g\left(\hat{B}\right) \right] = \tau \left[ f(A, B) L_g(B) \right], \tag{29}$$

for all bounded borel $f$. This is Proposition 1, proved below using operator-valued free probability.

Combining these, we have $\left\langle f(a) g\left(\hat{b}\right) \right\rangle_{(a,\hat{b}) \sim \mu_{A,B}} = \tau \left[ f(A) L_g(B) \right]$. Writing the right hand side as a scalar expectation and then conditioning on $b$,

$$\left\langle f(a) g\left(\hat{b}\right) \right\rangle_{(a,\hat{b}) \sim \mu_{A,B}} = \left\langle \langle f(a) \rangle_{a \sim \mu_{A|B=b}} L_g(b) \right\rangle_{b \sim \mu_B}$$

$$= \langle Q(b) L_g(b) \rangle_{b \sim \mu_B},$$

where $Q(b) := \langle f(a) \rangle_{a \sim \mu_{A|B=b}}$. Applying (28) and (29) again,

$$\left\langle f(a) g\left(\hat{b}\right) \right\rangle_{(a,\hat{b}) \sim \mu_{A,B}} = \tau \left[ Q(B) g\left(\hat{B}\right) \right]$$

$$= \left\langle Q(b) g\left(\hat{b}\right) \right\rangle_{(b,\hat{b}) \sim \mu_{B,\hat{B}}}$$

$$= \left\langle \langle f(a) \rangle_{a \sim \mu_{A|B=b}} \langle g(\hat{b}) \rangle_{\hat{b} \sim \mu_{\hat{B}|B=b}} \right\rangle_{b \sim \mu_B},$$

which shows that $a, \hat{b}$ are independent conditioned on $b$:

$$\mu_{A,\hat{B}} = \int \mu_{A|B=b} \mu_{\hat{B}|B=b} d\mu_B(b).$$

Applying, for example, classical $\epsilon$-gaussian smoothing to the measures of $A, B, \hat{B}$, we can assume that $\mu_{X,Y} \ll \mu_X \otimes \mu_Y$ for any two of the three. Thus we are free to form the Radon-Nikodym derivative $\frac{d\mu_{X,Y}}{d\mu_X \otimes \mu_Y}$, which corresponds to the gaussian-smoothed overlap function $O_{X,Y;\epsilon}(x, y)$. Since $O_{X,Y;\epsilon}(x, y) \mu_X(dx) = \mu_{X|Y=y}(dx)$, we have that for any bounded measurable function

$$\int \phi\left(a, \hat{b}\right) d\mu_{A,\hat{B}} = \iint \phi\left(a, \hat{b}\right) \left( \int \mu_{A|B=b}(da) \mu_{\hat{B}|B=b}\left(d\hat{b}\right) \mu_B(db) \right)$$

$$= \iint \phi\left(a, \hat{b}\right) \int \left( O_{A,B;\epsilon}(a, b) \mu_A(da) \right) \left( O_{B,\hat{B};\epsilon}\left(b, \hat{b}\right) \mu_{\hat{B}}\left(d\hat{b}\right) \right) \mu_B(db)$$

$$= \iint \phi\left(a, \hat{b}\right) \left( \int O_{A,B;\epsilon}(a, b) O_{B,\hat{B};\epsilon}\left(b, \hat{b}\right) \mu_B(db) \right) \mu_A(da) \mu_{\hat{B}}\left(d\hat{b}\right),$$

so the last quantity is exactly the Radon-Nikodym density $O_{A,\hat{B};\epsilon}$:

$$O_{A,\hat{B};\epsilon}\left(a, \hat{b}\right) = \int O_{A,B;\epsilon}(a, b) O_{B,\hat{B};\epsilon}\left(b, \hat{b}\right) \mu_B(db).$$

Taking the smoothing to 0, one obtains the transfer law for the overlap function. Note that one may have to interpret the $O_{X,Y}$ as distributions (eg involving $\delta$ kernels) in case of degenerate overlap between two of the matrices (eg. $A = B$).

**Proposition 1.** *Let $A, B$ be free from $X$, and let $\hat{B} = F(B, X)$ be a rational function of $B, X$. Then there is a linear operator $L$ on functions such that for arbitrary bounded borel $H$,*

$$\tau \left[ H(A, B) g\left(\hat{B}\right) \right] = \tau \left[ H(A, B) L[g](B) \right].$$

*Proof.* Let $\mathbf{B}$ be a $k \times k$ linearization of $\hat{B}$ such that $\left[ \mathbf{B}^{-1} \right]_{k,1} = g(F(B, X))$. Decompose $\mathbf{B}$ into its $X$-dependent and $X$-independent parts:

$$\mathbf{B} = X\mathbf{B}_X + \mathbf{B}_0.$$

Now form the linearization matrix

$$\mathbf{L}(M) := \begin{pmatrix} -\mathbf{B} & 0 \\ M\mathbf{e}_k^\top & -1 \end{pmatrix},$$

so that

$$\mathbf{L}^{-1}(M) := \begin{pmatrix} -\mathbf{B}^{-1} & 0 \\ -M\mathbf{e}_k^\top \mathbf{B}^{-1} & -1 \end{pmatrix}.$$

In particular, $\mathbf{L}_{k,1}^{-1}(M) = -g(F(B, X))$ and $\mathbf{L}_{k+1,1}^{-1}(M) = -Mg(F(B, X))$. So now

$$\left[ g_{\mathbf{L}(H(A,B))}(0) \right]_{k+1,1} = \tau \left[ \left[ (-\mathbf{L}(H(A,B)))^{-1} \right]_{k+1,1} \right]$$
$$= \tau \left[ H(A, B) g(F(B, X)) \right].$$

As with $\mathbf{B}$, decompose $\mathbf{L}$ into $X$-dependent and $X$-independent parts:

$$\mathbf{L}(M) = X\mathbf{L}_X + \mathbf{L}_0.$$

Assuming $M$ is a function of $A, B$ only, these two parts are operator-free (ie. with amalgamation over the space of $(k+1) \times (k+1)$ matrices with complex entries). By the additive subordination law,

$$g_{\mathbf{L}(M)}(0) = g_{\mathbf{L}_0}\left( -\mathcal{R}_{X\mathbf{L}_X}\left( g_{\mathbf{L}(M)}(0) \right) \right).$$

From the linearization, $g_{\mathbf{L}(M)}(0)$ is block lower triangular, and $X\mathbf{L}_X$ only has nonzero components in the upper left block. This implies that $\mathcal{R}_{X\mathbf{L}_X}\left( g_{\mathbf{L}(M)}(0) \right)$ is also confined to the upper left block, and that this entry is simply $\mathcal{R}_{-X\mathbf{B}_X}\left( \mathcal{E}\mathbf{B}^{-1} \right)$, where $\mathcal{E} := \tau \otimes \mathrm{id}$ is the expectation functional of the operator-space. These imply

$$\tau \left[ H(A, B) g(F(B, X)) \right] = \left[ g_{\mathbf{L}(H(A,B))}(0) \right]_{k+1,1}$$
$$= \left[ g_{\mathbf{L}_0}\left( -\mathcal{R}_{X\mathbf{L}_X}\left( g_{\mathbf{L}(H(A,B))}(0) \right) \right) \right]_{k+1,1}$$
$$= \tau \left[ \begin{pmatrix} \mathbf{B}_0 - \mathcal{R}_{-X\mathbf{B}_X}\left( \mathcal{E}\mathbf{B}^{-1} \right) & 0 \\ -H(A, B)\mathbf{e}_k^\top & 1 \end{pmatrix}^{-1} \right]_{k+1,1}$$
$$= \tau \left[ H(A, B) \left( \mathbf{B}_0 - \mathcal{R}_{-X\mathbf{B}_X}\left( \mathcal{E}\mathbf{B}^{-1} \right) \right)_{k,1}^{-1} \right].$$

The second factor can be regarded simply as a function of $B$ ($\mathbf{B}_0$ is the non-$X$ part of $\mathbf{B}$ and $\mathcal{E}\mathbf{B}^{-1} : B \mapsto \mathcal{E}\mathbf{B}^{-1}(B, X) \in M_k(\mathbb{C})$. This proves that there is some operator $L[g]$ as in the statement of the proposition. $L$ must obviously be linear in $g$, completing the proof. $\square$

## C  TWO-LOSS GEOMETRY IN ANISOTROPIC RIDGE REGRESSION

In this appendix we derive equation (11), prove Theorem 3, and obtain formulas for the loss and overlap functions that are used in figures 1-3. We consider ridge regression on multivariate gaussian input data with train and test covariances $\Sigma_{\text{train}}, \Sigma_{\text{test}}$ and with linear ground truth, $y(x) = \frac{1}{\sqrt{d}} w_* \cdot x$. We will assume for simplicity that $w_* \sim \mathcal{N}(0, I_d)$. A noisy training set is generated by sampling inputs as follows. The training set consists of

$$x_i \sim \mathcal{N}(0, \Sigma_{\text{train}}), \quad y_i = y(x_i) + \xi_i, \quad \xi_i \sim \mathcal{N}(0, \sigma_\xi^2),$$

for $i = 1, \ldots, m$. We define the sampling density $\alpha := m/d$. It will occasionally be convenient to state formulas in terms of $\alpha$'s reciprocal, $q := d/m$.

Using the framing of section 3.1.1, we will regard the noise $\xi$ as perturbing a noiseless training objective. The train and test losses are formulated as follows:

$$\mathcal{L}_{\text{train}}(w, \xi) := \tfrac{1}{2} \left[ \frac{1}{m} \sum_{i=1}^{m} \left( y_i - \tfrac{1}{\sqrt{d}} w \cdot x_i \right)^2 \right] + \tfrac{\lambda}{2d} \|w\|^2$$

$$\mathcal{L}_{\text{test}}(w) := \tfrac{1}{2} \left[ \mathbb{E}_x \left( y(x) - \tfrac{1}{\sqrt{d}} w \cdot x \right)^2 \right].$$

Note 1) we keep $\mathcal{L}_{\text{train}}$'s dependence on the perturbation $\xi$ explicit, and 2) the noise is not included in the test loss (and when it is, after averaging, it changes the loss only by an additive constant). Finally, let us write $\hat{w}(\xi) := \text{argmin}_w \mathcal{L}_{\text{train}}(w, \xi)$ for the learned weights, $\hat{y}(x) := \tfrac{1}{\sqrt{d}} \hat{w} \cdot x$ for the learned model, and $H_{\text{train}} := d \nabla^2 \mathcal{L}_{\text{train}}$ and $H_{\text{test}} := d \nabla^2 \mathcal{L}_{\text{test}}$ for the train and test Hessians; these scalings are chosen to keep spectra $O(1)$.

We begin by deriving equation (11) by applying the fluctuation law (6). To do so, we first compute $z, H_{\text{train}}, H_{\text{test}}$, and $C_{\text{train}}$. Differentiating, we find

$$\nabla_w \mathcal{L}_{\text{train}}(w, \xi) = \tfrac{1}{d} \left( \tfrac{1}{m} X^\top X + \lambda I \right) w - \tfrac{1}{\sqrt{d}} \tfrac{1}{m} X^\top \left( \tfrac{1}{\sqrt{d}} X w_* + \xi \right) \tag{30}$$

$$H_{\text{train}} := d \nabla_w^2 \mathcal{L}_{\text{train}} = \tfrac{1}{m} X^\top X + \lambda I. \tag{31}$$

Similarly,

$$\mathcal{L}_{\text{test}}(w) = \tfrac{1}{2d} (w - w_*) \Sigma_{\text{test}} (w - w_*).$$

$$H_{\text{test}} := d \nabla_w^2 \mathcal{L}_{\text{test}} = \Sigma_{\text{test}}.$$

Next, $z := d \nabla_w \mathcal{L}_{\text{train}}(w_0, \xi)$ is the scaled train gradient evaluated at the unperturbed optimum $w_0$. Since, by definition, $\nabla_w \mathcal{L}_{\text{train}}(w_0, 0) = 0$, substituting into (30) gives

$$z = d \nabla_w \mathcal{L}_{\text{train}}(w_0, \xi) = \left( \tfrac{1}{m} X^\top X + \lambda I \right) w_0 - \tfrac{\sqrt{d}}{m} X^\top \left( \tfrac{1}{\sqrt{d}} X w_* + \xi \right) \tag{32}$$

$$= -\tfrac{\sqrt{d}}{m} X^\top \xi. \tag{33}$$

Finally,

$$\begin{aligned} C_{\text{train}} &= \mathbb{E}_\xi \left[ (H_{\text{train}}^{-1} z)(H_{\text{train}}^{-1} z)^\top \right] \\ &= \tfrac{d}{m^2} \mathbb{E}_\xi \left[ H_{\text{train}}^{-1} X^\top \xi \xi^\top X H_{\text{train}}^{-1} \right] \\ &= \sigma_\xi^2 \tfrac{d}{m} H_{\text{train}}^{-1} (\tfrac{1}{m} X^\top X) H_{\text{train}}^{-1} \\ &= \sigma_\xi^2 \alpha^{-1} \hat{\Sigma}_{\text{train}} (\hat{\Sigma}_{\text{train}} + \lambda I)^{-2}, \end{aligned}$$

where $\hat{\Sigma}_{\text{train}} := \tfrac{1}{m} X^\top X$ is the (uncentered) train covariance. Summarizing,

$$z = -\tfrac{\sqrt{d}}{m} X^\top \xi \tag{34}$$

$$H_{\text{train}} = \hat{\Sigma}_{\text{train}} + \lambda I \tag{35}$$

$$H_{\text{test}} = \Sigma_{\text{test}} \tag{36}$$

$$C_{\text{train}} = \sigma_\xi^2 \alpha^{-1} \hat{\Sigma}_{\text{train}} (\hat{\Sigma}_{\text{train}} + \lambda I)^{-2}. \tag{37}$$

Since $\mathbb{E}[\xi] = 0$, we have $\mathbb{E}[\Delta w] = -\mathbb{E}[H_{\text{train}}^{-1} z] = 0$. Plugging directly into (6), we find

$$\mathbb{E}[\Delta \mathcal{L}] = \tfrac{1}{2} \iint \lambda_1 \lambda_2 \, O_{H_{\text{test}}, C_{\text{train}}}(\lambda_1, \lambda_2) \, \mu_{H_{\text{test}}}(d\lambda_1) \, \mu_{C_{\text{train}}}(d\lambda_2). \tag{38}$$

Since $\hat{\Sigma}_{\text{train}}, C_{\text{train}}$ commute, they share eigenvectors and we are free to replace $O_{H_{\text{test}}, C_{\text{train}}}$ with $O_{H_{\text{test}}, \hat{\Sigma}_{\text{train}}}$. Replacing the integral over $C_{\text{train}}$'s spectrum with one over $\hat{\Sigma}_{\text{train}}$, we find

$$\mathbb{E}[\Delta \mathcal{L}] = \tfrac{\sigma_\xi^2}{2\alpha} \iint \lambda_1 \frac{\lambda_2}{(\lambda_2 + \lambda)^2} O_{\Sigma_{\text{test}}, \hat{\Sigma}_{\text{train}}}(\lambda_1, \lambda_2) \, \mu_{\Sigma_{\text{test}}}(d\lambda_1) \, \mu_{\hat{\Sigma}_{\text{train}}}(d\lambda_2), \tag{39}$$

which is equation (11).

High-dimensional ridge regression has been studied extensively, so rather than rederiving published formulas, as much as possible, we restrict attention to the novel focus of this article: overlap decompositions. We treat the label noise as a fluctuation of the training objective, and derive exact asymptotic formulas for the needed train-test spectra and overlap functions that describing the effect of the fluctuation on test error.

All formulas are obtainable from the general trace formula stated in the following two propositions, which we prove in Appendix D.

**Proposition 2.** *The equation*

$$r = \left(1 - q \int \frac{t}{z - tr} d\mu_{\Sigma_{\text{train}}}(t)\right)^{-1}, \tag{40}$$

*has a unique solution $r$ satisfying $r \in \mathbb{H}^{\pm}$ for $z \in \mathbb{H}^{\mp}$ and satisfying $0 < r < 1$ for $z < 0$. This defines a holomorphic function $r(z)$ on all of $\mathbb{C}\backslash\mathbb{R}^{\geq 0}$ that is obtainable for each $z$ by fixed point iteration of the right hand side of* (40) *from an arbitrary initial point $r_0$ satisfying $r_0 \in \mathbb{H}^{\pm}$ for $z \in \mathbb{H}^{\mp}$ and $0 < r_0 < 1$ for $z < 0$.*

**Proposition 3.** *Let*

$$t_f(z) := \bar{tr}\left[f(\Sigma_{\text{test}}, \Sigma_{\text{train}})(zI - A)^{-1}\right], \tag{41}$$

*for bounded function $f$ and complex scalar $z \in \mathbb{C}\backslash\mathbb{R}^{\geq 0}$. As $m, d \to \infty$ with $q = d/m$ fixed,*

$$t_f(z) \to \bar{tr}\left[f(\Sigma_{\text{test}}, \Sigma_{\text{train}})(zI - r(z)\Sigma_{\text{train}})^{-1}\right],$$

*where $r(z)$ is the solution of the self-consistent equation* (40).

Thus to perform the calculation we simply express all quantities in terms of traces of the form $t_f(z)$, and then apply Propositions 2,3.

## C.1 TRAIN-TEST HESSIAN OVERLAP FUNCTION

Since we will be interested primarily in the ridgeless limit $\lambda \to 0$, and since the effect of nonzero $\lambda$ is simply to shift the spectrum of $H_{\text{train}} := \frac{1}{m}X^\top X + \lambda I$, we will omit $\lambda$ in the computation of $O_{H_{\text{test}}, H_{\text{train}}}(\lambda_1, \lambda_2)$, and will write $H_{\text{train}} = A = \frac{1}{m}X^\top X$ from now on unless explicitly stated otherwise. Thus, we are interested in the overlap function of the matrices

$$H_{\text{test}} = \Sigma_{\text{test}}, \qquad H_{\text{train}} = \Sigma_{\text{train}}^{1/2}\frac{1}{m}Z^\top Z\Sigma_{\text{train}}^{1/2}.$$

The computation is simplified significantly by taking advantage of asymptotic freeness of $\frac{1}{m}Z^\top Z$ from $\Sigma_{\text{train}}, \Sigma_{\text{test}}$. By Theorem 2, we have asymptotically

$$O_{H_{\text{test}}, H_{\text{train}}}(\lambda_{\text{te}}, \lambda_{\text{tr}}) = O_{\Sigma_{\text{test}}, H_{\text{train}}}(\lambda_{\text{te}}, \lambda_{\text{tr}})$$

$$= \int O_{\Sigma_{\text{test}}, \Sigma_{\text{train}}}(\lambda_{\text{te}}, \lambda) O_{\Sigma_{\text{train}}, H_{\text{train}}}(\lambda, \lambda_{\text{tr}}) d\mu_{\Sigma_{\text{train}}}(\lambda). \tag{42}$$

In particular this shows, somewhat intuitively, that the overlap function of the train/test Hessians will itself depend on the overlap function of the population covariance matrices. Eq. (42) shows the dependence is quite simple: One simply composes the overlap kernels taking an $H_{\text{train}} = \Sigma_{\text{train}}^{1/2}\frac{1}{m}Z^\top Z\Sigma_{\text{train}}^{1/2}$ eigenspace to a $\Sigma_{\text{train}}$ one, and taking a $\Sigma_{\text{train}}$ eigenspace to a $\Sigma_{\text{test}}$ one. The overlap of the train and test population covariances, $O_{\Sigma_{\text{test}}, \Sigma_{\text{train}}}$, is part of the input data of the problem and is therefore known. As for the second factor, $O_{\Sigma_{\text{train}}, H_{\text{train}}}$, this is simply the overlap function of the population and sample covariance matrices for an anisotropic gaussian sample. Formulas for this quantity are known (see, eg. Potters & Bouchaud (2020)). To keep the presentation self-contained we quickly derive an expression using operator-valued free probability.

Following Appendix B.1, the train-test Hessian overlap function can be computed via

$$O_{\Sigma_{\text{train}}, H_{\text{train}}}(\lambda_1, \lambda_2) := \lim_{\sigma \to 0} \frac{\bar{tr}\left[K(\Sigma_{\text{train}}; \lambda_1, \sigma_1) K(H_{\text{train}}; \lambda_2, \sigma_2)\right]}{\bar{tr}\left[K(\Sigma_{\text{train}}; \lambda_1, \sigma_1)\right]\bar{tr}\left[K(H_{\text{train}}; \lambda_2, \sigma_2)\right]},$$

where

$$K(x; \mu, \sigma) := \frac{1}{\pi} \frac{\sigma}{(x - \mu)^2 + \sigma^2},$$

is the Poisson kernel with center $\mu$ and width $\sigma$. Moving the first denominator trace into the numerator and canceling a factor of $\pi$ from top and bottom, we find that computing $O_{H_{\text{test}}, H_{\text{train}}}(\lambda_1, \lambda_2)$ requires the numerator and denominator traces

$$\bar{\text{tr}}\left[h(\Sigma_{\text{train}}) \frac{\sigma_2}{(A - \lambda_2 I)^2 + \sigma_2^2 I}\right], \qquad \bar{\text{tr}}\left[\frac{\sigma_2}{(A - \lambda_2 I)^2 + \sigma_2^2 I}\right],$$

where $h(\Sigma_{\text{train}}) := K(\Sigma_{\text{train}}; \lambda_1, \sigma_1) / \bar{\text{tr}}[K(\Sigma_{\text{train}}; \lambda_1, \sigma_1)]$.

Using the definition of $t_f(z)$ (41), and the resolvent form of the Poisson kernel,

$$K(x; \mu, \sigma) = -\frac{1}{\pi} \text{Im}\,(\mu + i\sigma - x)^{-1},$$

these become

$$-\text{Im}\, t_h(\lambda_2 + i\sigma_2), \qquad -\text{Im}\, t_1(\lambda_2 + i\sigma_2).$$

Proposition 3 implies

$$t_h(z) \to \bar{\text{tr}}\left[h(\Sigma_{\text{train}})(zI - r(z)\Sigma_{\text{train}})^{-1}\right] \tag{43}$$

$$t_1(z) \to \bar{\text{tr}}\left[(zI - r(z)\Sigma_{\text{train}})^{-1}\right], \tag{44}$$

so

$$O_{\Sigma_{\text{train}}, H_{\text{train}}}(\lambda_1, \lambda_2) \to \lim_{\sigma_1 \to 0} \lim_{z \to \lambda_2^{+i}} \frac{\text{Im}\, \bar{\text{tr}}\left[h(\Sigma_{\text{train}})(zI - r(z)\Sigma_{\text{train}})^{-1}\right]}{\text{Im}\, \bar{\text{tr}}\left[(zI - r(z)\Sigma_{\text{train}})^{-1}\right]},$$

where $\lim_{z \to \lambda_2^{+i}}$ is shorthand for $\lim_{\sigma \to 0}$ with $z = \lambda_2 + i\sigma$. Taking $\sigma_1 \to 0$ sends $h(\Sigma_{\text{train}})$ to a delta function and collapses the trace in the numerator to the $\lambda_1$ eigenspace of $\Sigma_{\text{train}}$, so

$$O_{\Sigma_{\text{train}}, H_{\text{train}}}(\lambda_1, \lambda_2) \to \lim_{z \to \lambda_2^{+i}} \frac{\text{Im}\, \frac{1}{z - r(z)\lambda_1}}{\int \text{Im}\, \frac{1}{z - r(z)\lambda} d\mu_{\Sigma_{\text{train}}}(\lambda)}.$$

Composing with $O_{\Sigma_{\text{test}}, \Sigma_{\text{train}}}$ yields the overlap function $O_{\Sigma_{\text{test}}, H_{\text{train}}} = O_{H_{\text{test}}, H_{\text{train}}}$.

## C.2 Overlap decomposition of $\Delta\mathcal{L}$

Trace integrals are written in terms of the spectra and overlaps of the matrices involved. To explicitly determine the spectral density of $H_{\text{train}}$, note that it can be written in terms of the trace in equation (44),

$$\rho_{H_{\text{train}}}(\lambda_{\text{tr}}) = \lim_{\sigma \to 0} \bar{\text{tr}}[K(H_{\text{train}}; \lambda_{\text{tr}}, \sigma)].$$

Using the same approach as above, we have the following for the $\sigma$-Poisson-smoothed spectral density of $H_{\text{train}}$:

$$\rho_{H_{\text{train}}; \sigma}(\lambda_{\text{tr}}) = -\frac{1}{\pi} \text{Im}\, t_1(\lambda_{\text{tr}} + i\sigma)$$

$$\to -\frac{1}{\pi} \int \text{Im}\, \frac{1}{z - r(z)\lambda} d\mu_{\Sigma_{\text{train}}}(\lambda).$$

Collecting the results of the previous section and the fluctuation formula (11),

$$\Delta\mathcal{L} = \frac{\sigma^2}{2\alpha} \iint \underbrace{\lambda_{\text{te}}}_{\text{test curvature}} \underbrace{\frac{\lambda_{\text{tr}}}{(\lambda_{\text{tr}} + \lambda)^2}}_{\text{train variance}} \underbrace{O_{H_{\text{train}}, H_{\text{test}}}(\lambda_{\text{te}}, \lambda_{\text{tr}})}_{\text{eigenspace overlap}} \mu_{H_{\text{test}}}(d\lambda_{\text{te}})\, \mu_{H_{\text{train}}}(d\lambda_{\text{tr}}), \tag{45}$$

where

$$O_{H_{\text{test}}, H_{\text{train}}}(\lambda_{\text{te}}, \lambda_{\text{tr}}) = \int O_{\Sigma_{\text{test}}, \Sigma_{\text{train}}}(\lambda_{\text{te}}, \lambda)\, O_{\Sigma_{\text{train}}, H_{\text{train}}}(\lambda, \lambda_{\text{tr}})\, d\mu_{\Sigma_{\text{train}}}(\lambda), \tag{46}$$

with

$$O_{\Sigma_{\text{train}}, H_{\text{train}}}(\lambda_1, \lambda_2) \to \lim_{z \to \lambda_2^{+i}} \frac{\text{Im} \frac{1}{z - r(z)\lambda_1}}{\int \text{Im} \frac{1}{z - r(z)\lambda} d\mu_{\Sigma_{\text{train}}}(\lambda)}.$$

This provides a complete decomposition of the test loss fluctuation in terms of spectra and overlaps of the train and test Hessian.

### C.3 Proof of covariate shift theorem 3

This subsection, together with the proofs of Propositions 2, 3 found in Appendix D, proves Theorem 3.

Formulas (45) and (46) show the effect of covariate shift in train/test sets decomposes naturally in terms of the overlap function $O_{\Sigma_{\text{test}}, \Sigma_{\text{train}}}$ of the two population covariances. (Note that there are two levels of overlap decomposition: the test loss increment is decomposed in terms of the train-test Hessian overlap function (45), which in turn is decomposed in terms of the overlaps of $\Sigma_{\text{test}}, \Sigma_{\text{train}}$.)

We can equivalently start from explicit expressions for the fluctuation. Differentiating the loss and solving for the optimal weights directly, one has

$$\mathcal{L}_{\text{test}}(\hat{w}(\xi)) = -\frac{1}{2} \left( \frac{1}{\alpha} \sigma_\xi^2 (t_{\text{id}}(-\lambda) - \lambda t'_{\text{id}}(-\lambda)) + \lambda^2 t'_{\text{id}}(-\lambda) \right), \tag{47}$$

(equation (51) of Appendix C.4). Since $\Delta\mathcal{L} = \mathcal{L}_{\text{test}} - \mathcal{L}_0$, and $\mathcal{L}_0$ is obtained by simply setting the perturbation strength $\sigma_\xi \to 0$, we immediately find

$$\Delta\mathcal{L} = -\frac{\sigma_\xi^2}{2\alpha} \frac{d}{d\lambda} \lambda t_{\text{id}}(-\lambda). \tag{48}$$

Adopting the notation $\tilde{\lambda} := \frac{\lambda}{r(-\lambda)}$, Proposition 3 yields

$$\lambda t_{\text{id}}(-\lambda) \to -\tilde{\lambda} \bar{\text{tr}} \left[ \Sigma_{\text{test}} \left( \tilde{\lambda} I + \Sigma_{\text{train}} \right)^{-1} \right]$$

$$\frac{d}{d\lambda} \lambda t_{\text{id}}(-\lambda) \to -\tilde{\lambda}' \bar{\text{tr}} \left[ \Sigma_{\text{test}} \Sigma_{\text{train}} \left( \tilde{\lambda} I + \Sigma_{\text{train}} \right)^{-2} \right].$$

Substituting into (48),

$$\Delta\mathcal{L} \to \frac{\sigma_\xi^2}{2\alpha} \tilde{\lambda}' \bar{\text{tr}} \left[ \Sigma_{\text{test}} \Sigma_{\text{train}} \left( \tilde{\lambda} I + \Sigma_{\text{train}} \right)^{-2} \right].$$

Writing the last trace as an integral over the spectral measures of $\Sigma_{\text{test}}, \Sigma_{\text{train}}$, this becomes

$$\Delta\mathcal{L} \to \frac{\sigma_\xi^2}{2\alpha} \tilde{\lambda}' \int \lambda_{\text{te}} \frac{\lambda_{\text{tr}}}{\left( \tilde{\lambda} + \lambda_{\text{tr}} \right)^2} O_{\Sigma_{\text{test}}, \Sigma_{\text{train}}}(\lambda_{\text{te}}, \lambda_{\text{tr}}) d\mu_{\Sigma_{\text{test}}}(\lambda_{\text{te}}) d\mu_{\Sigma_{\text{train}}}(\lambda_{\text{tr}}), \tag{49}$$

which completes the proof of Theorem 3. Equation (49) parallels (45) but averages out the random training inputs and label noise to express $\Delta\mathcal{L}$ purely in terms of the known objects $\Sigma_{\text{train}}, \Sigma_{\text{test}}$. This expression shows that label noise leads to large increases in test loss when a direction of large training variance (small eigenvalue $\lambda_{\text{tr}}$ of $\Sigma_{\text{train}}$) and a direction of large test curvature (large eigenvalue $\lambda_{\text{te}}$ of $\Sigma_{\text{test}}$) experience significant overlap (large $O_{\Sigma_{\text{test}}, \Sigma_{\text{train}}}(\lambda_{\text{te}}, \lambda_{\text{tr}})$).

### C.4 Explicit formulas for test loss, fluctuation

Here we derive explicit expressions for the full test loss and test loss fluctuation under general covariate shift. Since these formulas and generalizations of them are already published, this section is mostly for internal reference—especially for calculation of theoretical loss curves in Figs. 1 and 2.

Let $X$ have rows $x_i^\top$ and $\xi$ have components $\xi_i$. $\mathcal{L}_{\text{train}}$ can be written

$$\mathcal{L}_{\text{train}}(w, \xi) := \frac{1}{2m} \| \frac{1}{\sqrt{d}} X w_* + \xi - \frac{1}{\sqrt{d}} X w \|^2 + \frac{\lambda}{2d} w^\top w.$$

Differentiating, we find

$$\nabla \mathcal{L}_{\text{train}}(w, \xi) = \frac{1}{d} H_{\text{train}} w - \frac{1}{\sqrt{d}} \frac{1}{m} X^\top \left( \frac{1}{\sqrt{d}} X w_* + \xi \right)$$

$$H_{\text{train}} := d \nabla^2 \mathcal{L}_{\text{train}} = \frac{1}{m} X^\top X + \lambda I.$$

Similarly,

$$\mathcal{L}_{\text{test}}(w) = \frac{1}{2d} (w - w_*) \Sigma_{\text{test}} (w - w_*).$$

$$H_{\text{test}} := d \nabla^2 \mathcal{L}_{\text{test}} = \Sigma_{\text{test}}.$$

Solving $0 = \nabla_w \mathcal{L}_{\text{train}}$ yields

$$\hat{w} = H_{\text{train}}^{-1} \left( \frac{1}{m} X^\top X \right) w_* + (H_{\text{train}})^{-1} \frac{\sqrt{d}}{m} X^\top \xi.$$

Substituting into $\mathcal{L}_{\text{test}}$ yields

$$\mathcal{L}_{\text{test}}(\hat{w}(\xi)) = \frac{1}{2} \bar{\text{tr}} \left[ \Sigma_{\text{test}} \frac{q \sigma_\xi^2 \frac{1}{m} X^\top X + \lambda^2 I}{\left( \frac{1}{m} X^\top X + \lambda I \right)^2} \right].$$

Since $(A + \lambda I)^{-2} = -\frac{d}{d\lambda} (A + \lambda I)^{-1}$, we can write

$$\mathcal{L}_{\text{test}}(\hat{w}(\xi)) = -\frac{1}{2} \left( q \sigma_\xi^2 \frac{d}{d\lambda} \lambda - \lambda^2 \frac{d}{d\lambda} \right) t_{\text{id}}(-\lambda) \tag{50}$$

$$= -\frac{1}{2} \left( q \sigma_\xi^2 (t_{\text{id}}(-\lambda) - \lambda t'_{\text{id}}(-\lambda)) + \lambda^2 t'_{\text{id}}(-\lambda) \right) \tag{51}$$

Proposition 3 implies

$$t_{\text{id}}(z) \to \bar{\text{tr}} \left[ \Sigma_{\text{test}} (z I - r(z) \Sigma_{\text{train}})^{-1} \right]$$

$$t'_{\text{id}}(z) \to -\bar{\text{tr}} \left[ \Sigma_{\text{test}} (z I - r(z) \Sigma_{\text{train}})^{-2} (I - r'(z) \Sigma_{\text{train}}) \right],$$

which fully specifies $\mathcal{L}_{\text{test}}(\hat{w}(\xi))$. The fluctuation is easily gotten by setting $\sigma_\xi^2 \to 0$ and subtracting from $\mathcal{L}_{\text{test}}(\hat{w}(\xi))$.

**Reduction to published formulas** Letting $\tilde{\lambda} := \frac{\lambda}{r(-\lambda)}$ and substituting into (40), we obtain

$$\lambda = \tilde{\lambda} - \frac{1}{\alpha} \int \frac{\tilde{\lambda} t}{\tilde{\lambda} + t} d\mu_{\Sigma_{\text{train}}}(t), \tag{52}$$

which is eq. (8) of Mel & Ganguli (2021) for the "effective regularization".

The fluctuation in (50) is

$$\Delta \mathcal{L} = -q \sigma_\xi^2 \frac{1}{2} \frac{d}{d\lambda} \lambda t_{\text{id}}(-\lambda).$$

Since

$$\lambda t_{\text{id}}(-\lambda) \to \frac{\lambda}{r(-\lambda)} \bar{\text{tr}} \left[ \Sigma_{\text{train}} \left( -\frac{\lambda}{r(-\lambda)} I - \Sigma_{\text{train}} \right)^{-1} \right]$$

$$= -\tilde{\lambda} \bar{\text{tr}} \left[ \Sigma_{\text{train}} \left( \tilde{\lambda} I + \Sigma_{\text{train}} \right)^{-1} \right],$$

we get

$$\Delta \mathcal{L} = q \sigma_\xi^2 \frac{1}{2} \tilde{\lambda}' \bar{\text{tr}} \left[ \left( \frac{\Sigma_{\text{train}}}{\tilde{\lambda} I + \Sigma_{\text{train}}} \right)^2 \right].$$

The authors define $\frac{1}{\rho_f} := \frac{d\tilde{\lambda}}{d\lambda}$, so

$$\Delta \mathcal{L} = \frac{1}{2} q \sigma_\xi^2 \frac{1}{\rho_f} \bar{\text{tr}} \left[ \left( \frac{\Sigma_{\text{train}}}{\tilde{\lambda} I + \Sigma_{\text{train}}} \right)^2 \right],$$

which matches the fluctuation term of their formula up to constant factors differing in the loss definitions. Next, the remaining term can be written

$$\mathcal{L}_0 = \mathcal{L}_{\text{test}} - \Delta\mathcal{L} = \frac{1}{2}\left(-\lambda t_{\text{id}}(-\lambda) + \lambda\frac{d}{d\lambda}\lambda t_{\text{id}}(-\lambda)\right).$$

Using (50) again,

$$\mathcal{L}_0 = \frac{1}{2}\left(\bar{\text{tr}}\left[\frac{\tilde{\lambda}\Sigma_{\text{train}}}{\tilde{\lambda}I + \Sigma_{\text{train}}}\right] - \lambda\tilde{\lambda}'\bar{\text{tr}}\left[\frac{\Sigma_{\text{train}}^2}{\left(\tilde{\lambda}I + \Sigma_{\text{train}}\right)^2}\right]\right)$$

Comparing to (52), the first term is $\alpha\left(\tilde{\lambda} - \lambda\right)$, and differentiating gives

$$\tilde{\lambda}' = \frac{\alpha}{\alpha - \text{tr}\left[\left(\frac{\Sigma_{\text{train}}}{\tilde{\lambda}+\Sigma_{\text{train}}}\right)^2\right]}.$$

Substituting and simplifying yields

$$\begin{aligned}\mathcal{L}_0 &= \frac{1}{2}\left(\alpha\tilde{\lambda} - \alpha\lambda\frac{\alpha}{\alpha - \text{tr}\left[\left(\frac{\Sigma_{\text{train}}}{\tilde{\lambda}+\Sigma_{\text{train}}}\right)^2\right]}\right)\\ &= \frac{1}{2}\tilde{\lambda}'\left(\tilde{\lambda}\left(\alpha - \text{tr}\left[\left(\frac{\Sigma_{\text{train}}}{\tilde{\lambda}+\Sigma_{\text{train}}}\right)^2\right]\right) - \alpha\lambda\right).\end{aligned}$$

Once again using equation (52), $\alpha\left(\tilde{\lambda} - \lambda\right)$ can be turned back into a trace:

$$\begin{aligned}\mathcal{L}_0 &= \frac{1}{2}\tilde{\lambda}'\left(\bar{\text{tr}}\left[\frac{\tilde{\lambda}\Sigma_{\text{train}}}{\tilde{\lambda}I + \Sigma_{\text{train}}}\right] - \tilde{\lambda}\text{tr}\left[\left(\frac{\Sigma_{\text{train}}}{\tilde{\lambda}+\Sigma_{\text{train}}}\right)^2\right]\right)\\ &= \frac{1}{2}\tilde{\lambda}'\bar{\text{tr}}\left[\frac{\tilde{\lambda}^2\Sigma_{\text{train}}}{\left(\tilde{\lambda}I + \Sigma_{\text{train}}\right)^2}\right],\end{aligned}$$

which is equivalent to their second term.

## C.5  $k$-LEVEL MODEL

At several points in the main text we refer to a $k$-level input covariance,

$$\mu_{\Sigma_{\text{train}}} = \sum_{i=1}^{k} p_i\delta_{s_i}.$$

In this case the self-consistent equation for $r$ (40) becomes

$$r = \left(1 - q\sum_{i=1}^{k}p_i\frac{s_i}{z - s_i r}\right)^{-1},$$

which can be written as $p(r, z) = 0$ for some polynomial in $r, z$. Similarly, the overlap function simplifies to a sum over the distinct eigenvalues of $\Sigma_{\text{train}}$:

$$O_{\Sigma_{\text{train}}, H_{\text{train}}}(\lambda_1, \lambda_2) \to \lim_{z\to\lambda_2^+ i}\frac{\text{Im}\frac{1}{z - r(z)\lambda_1}}{\sum_{i=1}^{k}p_i\text{Im}\frac{1}{z - r(z)s_i}}.$$

### C.5.1 SEPARATED SCALES LIMIT

We now assume the scales are widely separated: $s_{i+1} \ll s_i$. We will also work with the ridgeless formulas corresponding to $\lambda \to 0$ derived in Appendix C.6. For simplicity, assume $s_1 = 1$. We will obtain leading order formulas for $h(\alpha)$ as the ratio of successive scales is taken to $0$. In Appendix C.6, $h$ is defined and found to satisfy the following self-consistent equation (equation (55)):

$$1 = \int \frac{t}{h + \alpha t} d\mu_{\Sigma_{\text{train}}}(t)$$

The right hand side is a decreasing function of $h$ and a decreasing function of $\alpha$, we have that $h$ is a decreasing function of $\alpha$. Since the integral reduces to a sum over the $k$ eigenvalues, and since all terms with $t \ll h$ do not contribute at leading order in $s_{i+1}/s_i$, we assume that $h \approx s_1 = 1$ and neglect all lower terms, giving

$$1 = p_1 \frac{1}{h + \alpha},$$

so $h = p_1 - \alpha$ and $\Delta\mathcal{L}$ is

$$\Delta\mathcal{L} = \frac{\sigma_\xi^2}{2} \frac{\alpha \int sO(s,1) d\mu_{\Sigma_{\text{test}}}(s)}{p_1 - \alpha},$$

the integral in the numerator can be written

$$\int sO(s,1) d\mu_{\Sigma_{\text{test}}}(s) = \frac{1}{p_1} \bar{\text{tr}} \left[ \Sigma_{\text{test}} P_{\Sigma_{\text{train}}=1} \right],$$

where $P_{\Sigma_{\text{train}}=a}$ is the projector onto $\Sigma_{\text{train}}$'s $a$-eigenspace. In other words, the integral is simply the normalized total overlap of $\Sigma_{\text{test}}$ onto the strong training covariance space, and is equal to $1$ for $\Sigma_{\text{train}} = \Sigma_{\text{test}}$.

Now let us assume that $h$ is near the scale $s_i^2$. The self-consistent equation becomes

$$1 = p_i \frac{s_i^2}{h + \alpha s_i^2} + \frac{1}{\alpha} \mu_{\Sigma_{\text{train}}} \left( \gg s_i^2 \right),$$

where $\mu_{\Sigma_{\text{train}}} \left( \gg s_i^2 \right)$ is the total probability mass of all scales greater than $s_i^2$, ie $\sum_{j=1}^{i-1} p_j$. Solving yields

$$\alpha s_i^2 \left[ \frac{p_i}{\alpha - \mu_{\Sigma_{\text{train}}} \left( \gg s_i^2 \right)} - 1 \right] = h,$$

which is consistent with the assumption that $h \sim s_i^2$. Since $h \geq 0$, we only get a valid solution for $\alpha \geq \mu_{\Sigma_{\text{train}}} \left( \gg s_i^2 \right)$. Substituting back into the error expression yields

$$\Delta\mathcal{L} = \frac{\sigma_\xi^2}{2} \frac{p_i \alpha^2 \int \frac{st}{(h+\alpha t)^2} O(s,t) d\mu_{\Sigma_{\text{train}}}(t) d\mu_{\Sigma_{\text{test}}}(s)}{\left( \mu_{\Sigma_{\text{train}}} \left( \gg s_{i+1}^2 \right) - \alpha \right) \left( \alpha - \mu_{\Sigma_{\text{train}}} \left( \gg s_i^2 \right) \right)}.$$

Evaluating the numerator generally requires a choice of $\Sigma_{\text{test}}$'s behavior in the limit $s_{i+1}/s_i \to 0$, but note that the denominator has zeros at $\alpha = \mu_{\Sigma_{\text{train}}} \left( \gg s_i^2 \right), \mu_{\Sigma_{\text{train}}} \left( \gg s_{i+1}^2 \right)$, and so the error will generically become infinite whenever $\alpha$ is equal to the cumulative mass of some number of top scales. As a simple special case, letting $\Sigma_{\text{test}} = \Sigma_{\text{train}}$, this reduces to

$$\Delta\mathcal{L} = \frac{\sigma_\xi^2}{2} \frac{\left( \alpha - \mu_{\Sigma_{\text{train}}} \left( \gg s_i^2 \right) \right)^2 + p_i \mu_{\Sigma_{\text{train}}} \left( \gg s_i^2 \right)}{\left( \mu_{\Sigma_{\text{train}}} \left( \gg s_{i+1}^2 \right) - \alpha \right) \left( \alpha - \mu_{\Sigma_{\text{train}}} \left( \gg s_i^2 \right) \right)}.$$

Since under this assumption,

$$\mathcal{L}_0 = \frac{1}{2} h \, \bar{\text{tr}} \left[ \Sigma_{\text{train}} \left( hI + \alpha \Sigma_{\text{train}} \right)^{-1} \right] = \frac{1}{2} h$$

$$= \frac{1}{2} \left\{ \alpha s_i^2 \left[ \frac{p_i}{\alpha - \mu_{\Sigma_{\text{train}}} \left( \gg s_i^2 \right)} - 1 \right] \right\} \quad \mu_{\Sigma_{\text{train}}} \left( \gg s_i^2 \right) < \alpha \leq \mu_{\Sigma_{\text{train}}} \left( \gg s_{i+1}^2 \right),$$

when we take $s_{i+1}/s_i \to 0$, $\mathcal{L}_0$ only contributes at the highest scale, so

$$\mathcal{L}_0 = \frac{1}{2} \sigma_+ (p_1 - \alpha),$$

where $\sigma_+$ is the ReLU function.

## C.6 RIDGELESS LIMIT

Here we simplify our formula for the test error in the ridgeless limit. From (50), we have

$$\Delta\mathcal{L} = -\frac{1}{2}q\sigma_\xi^2 \frac{d}{d\lambda}\lambda t_{\text{id}}(-\lambda)$$

$$\mathcal{L}_0 := \mathcal{L}_{\text{test}} - \Delta\mathcal{L} = -\frac{1}{2}\lambda^2 t'_{\text{id}}(-\lambda).$$

It will also be helpful to consult $t_{\text{id}}$ and $r$'s explicit expressions as matrix traces (equations (41) and (59)):

$$t_{\text{id}}(z) = \bar{\text{tr}}\left[\Sigma_{\text{test}}\left(zI - \frac{1}{m}X^\top X\right)^{-1}\right], \tag{53}$$

$$r(z) = 1 + q\,\bar{\text{tr}}\left[(z - A)^{-1}A\right]. \tag{54}$$

### C.6.1 OVERSAMPLED REGIME

From (53), and since for $\alpha := m/d > 1$, the limiting spectrum of $\frac{1}{m}X^\top X$ is bounded away from 0, $t_{\text{id}}(-\lambda)$ is analytic as $\lambda \to 0^+$. Thus in the oversampled regime

$$\Delta\mathcal{L} \to -\frac{1}{2}q\sigma_\xi^2 t_{\text{id}}(0), \qquad \mathcal{L}_0 \to 0.$$

From Propositions 2 and 3,

$$t_{\text{id}}(z) \to -\frac{1}{r(0)}\bar{\text{tr}}\left[\Sigma_{\text{test}}\Sigma_{\text{train}}^{-1}\right], \qquad r(0) = 1 - q,$$

so that

$$\Delta\mathcal{L}, \mathcal{L}_{\text{test}} \to \frac{1}{2}\sigma_\xi^2 \frac{1}{\alpha - 1}\bar{\text{tr}}\left[\Sigma_{\text{test}}\Sigma_{\text{train}}^{-1}\right].$$

### C.6.2 UNDERSAMPLED REGIME

Now assume $\alpha < 1$. For $\lambda \to 0$, $t_{\text{id}}$ and $r$'s explicit expressions in (53) and (59) suggest $t_{\text{id}}(-\lambda) = O(\lambda^{-1})$ and $r(-\lambda) = O(\lambda)$. For convenience we will rewrite our formulas in terms of $h(z) := \frac{1}{q}\left(\frac{z}{r(-z)} - z\right)$. Substituting into the self-consistent equation for $r$ (40) and simplifying gives

$$h(z) = (qh(z) + z)\int \frac{t}{qh(z) + z + t}d\mu_{\Sigma_{\text{train}}}(t).$$

Now differentiating and setting $z \to 0$, we find

$$1 = q\int \frac{t}{qh + t}d\mu_{\Sigma_{\text{train}}}(t) \tag{55}$$

$$h' = \frac{\int \left(\frac{t}{qh+t}\right)^2 d\mu_{\Sigma_{\text{train}}}(t)}{1 - q\int \left(\frac{t}{qh+t}\right)^2 d\mu_{\Sigma_{\text{train}}}(t)}, \tag{56}$$

where we've suppressed the argument of $h, h'$.

We now write the error expressions in terms of these

$$\Delta\mathcal{L} = -\frac{1}{2}q\sigma_\xi^2 \frac{d}{d\lambda}\lambda t_{\text{id}}(-\lambda)$$

$$= \frac{1}{2}q\sigma_\xi^2 \frac{d}{d\lambda}(qh(\lambda) + \lambda)\bar{\text{tr}}\left[\Sigma_{\text{test}}\left((qh(\lambda) + \lambda)I + \Sigma_{\text{train}}\right)^{-1}\right]$$

$$\xrightarrow{\lambda \to 0} \frac{1}{2}q\sigma_\xi^2 \frac{\bar{\text{tr}}\left[\Sigma_{\text{test}}\frac{\Sigma_{\text{train}}}{(qhI + \Sigma_{\text{train}})^2}\right]}{1 - q\int \left(\frac{t}{qh+t}\right)^2 d\mu_{\Sigma_{\text{train}}}(t)}$$

$$= \frac{\sigma_\xi^2}{2}\frac{q\int \frac{st}{(qh+t)^2}O(s,t)d\mu_{\Sigma_{\text{train}}}(t)d\mu_{\Sigma_{\text{test}}}(s)}{1 - q\int \left(\frac{t}{qh+t}\right)^2 d\mu_{\Sigma_{\text{train}}}(t)},$$

while

$$\mathcal{L}_0 = -\frac{1}{2}\lambda^2 t'_{\mathrm{id}}\left(-\lambda\right)$$

$$= \frac{1}{2}qh\,\bar{\mathrm{tr}}\left[\Sigma_{\mathrm{test}}\left(qhI + \Sigma_{\mathrm{train}}\right)^{-1}\right]$$

$$= \frac{1}{2}qh\int\frac{s}{qh+t}O\left(s,t\right)d\mu_{\Sigma_{\mathrm{train}}}\left(t\right)d\mu_{\Sigma_{\mathrm{test}}}\left(s\right).$$

Finally, the total loss is just $\mathcal{L}_0 + \Delta\mathcal{L}$.

## D  CHARACTERIZATION OF $t_f\left(z\right)$

Here we derive an asymptotically exact expression for

$$t_F\left(z\right) := \bar{\mathrm{tr}}\left[F\left(\Sigma_{\mathrm{test}},\Sigma_{\mathrm{train}}\right)\left(zI - A\right)^{-1}\right].$$

Let us abbreviate $F := F\left(\Sigma_{\mathrm{train}},\Sigma_{\mathrm{test}}\right)$. First,

$$A = \frac{1}{m}X^\top X = \Sigma_{\mathrm{train}}^{1/2}\left(\frac{1}{m}Z^\top Z\right)\Sigma_{\mathrm{train}}^{1/2},$$

where $Z$ has standard normal entries, so that

$$t_F\left(z\right) = \bar{\mathrm{tr}}\left[\Sigma_{\mathrm{train}}^{1/2}F\Sigma_{\mathrm{train}}^{-1/2}\left(zI - \Sigma_{\mathrm{train}}\frac{1}{m}Z^\top Z\right)^{-1}\right]. \tag{57}$$

Now define $B := I_5 - E_{5,2}$, where $E_{5,2}$ is a matrix whose $(5,2)$ entry is 1 and has all other entries equal to 0, and let

$$\Sigma = \begin{pmatrix} 0 & 0 & 0 & 0 & 0 \\ \frac{1}{z}\Sigma_{\mathrm{train}}^{1/2}F\Sigma_{\mathrm{train}}^{-1/2} & 0 & \frac{1}{z}\Sigma_{\mathrm{train}} & 0 & 0 \\ 0 & 0 & 0 & 0 & 0 \\ 0 & 0 & 0 & 0 & 0 \\ 0 & 0 & 0 & 0 & 0 \end{pmatrix}$$

$$Q = \begin{pmatrix} 0 & 0 & 0 & 0 & 0 \\ 0 & 0 & 0 & 0 & 0 \\ 0 & 0 & 0 & \frac{1}{\sqrt{m}}Z^\top & 0 \\ 0 & 0 & 0 & 0 & \frac{1}{\sqrt{m}}Z \\ 0 & 0 & 0 & 0 & 0 \end{pmatrix}.$$

It is straightforward to verify that $\left(B - \left(\Sigma + Q\right)\right)^{-1}$ has as its $(5,1)$ block exactly the matrix in (57), and so $t_F\left(z\right) = \left[g_{\Sigma+Q}\left(B\right)\right]_{5,1}$, where $g_{\Sigma+Q}$ is the operator-valued Cauchy transform of $\Sigma + Q$.

By rotational invariance, $\Sigma, Q$ are asymptotically operator free, meaning we can apply the operator-valued additive subordination relation (see, eg. Mingo & Speicher (2017) Chapter 10), which yields the self-consistent equation

$$g_{\Sigma+Q}\left(B\right) = g_\Sigma\left(B - \mathcal{R}_Q\left(g_{\Sigma+Q}\left(B\right)\right)\right).$$

The blocks of $Q$ are standard normal matrices, and so its $\mathcal{R}$-transform is given by

$$\mathcal{R}_Q\left(M\right) = \mathcal{E}\left[QMQ\right],$$

where $\mathcal{E}$ is the operator-valued expectation that takes normalized traces of all square blocks. Due to the large number of zeros in $Q$, only two entries of $\mathcal{R}_Q\left(M\right)$ are nonzero:

$$\left[\mathcal{R}_Q\left(M\right)\right]_{3,5} = M_{4,4}$$

$$\left[\mathcal{R}_Q\left(M\right)\right]_{4,4} = qM_{5,3}.$$

On the other hand, by definition $g_\Sigma (M) = \mathcal{E} \left[ (M - \Sigma)^{-1} \right]$. Substituting back into the subordination relation and writing $g$ for $g_{\Sigma+Q} (B)$, we find

$$g = \mathcal{E} \left[ \begin{pmatrix} 1 & 0 & 0 & 0 & 0 \\ -\frac{1}{z}\Xi & 1 & -\frac{1}{z}\Sigma_{\text{train}} & 0 & 0 \\ 0 & 0 & 1 & 0 & -g_{44} \\ 0 & 0 & 0 & 1 - qg_{53} & 0 \\ 0 & -1 & 0 & 0 & 1 \end{pmatrix}^{-1} \right],$$

where to simplify notation we have written $\Xi := \Sigma_{\text{train}}^{1/2} F \Sigma_{\text{train}}^{-1/2}$. The entries of the right side are straightforward to compute using elementary row operations. Performing just enough such operations to determine the $(5,1)$, $(4,4)$, and $(5,3)$ entries, we obtain the closed system of equations

$$g_{53} = \bar{\text{tr}} \left[ \Sigma_{\text{train}} (zI - g_{44}\Sigma_{\text{train}})^{-1} \right]$$

$$g_{44} = \frac{1}{1 - qg_{53}}$$

$$g_{51} = \bar{\text{tr}} \left[ F (zI - g_{44}\Sigma_{\text{train}})^{-1} \right].$$

We can eliminate $g_{53}$ entirely, giving our trace

$$t_f (z) = g_{51} = \bar{\text{tr}} \left[ F (zI - r\Sigma_{\text{train}})^{-1} \right],$$

in terms of the scalar $r := g_{44}$ that satisfies

$$r = \left( 1 - q\bar{\text{tr}} \left[ \Sigma_{\text{train}} (zI - r\Sigma_{\text{train}})^{-1} \right] \right)^{-1}$$

A few remarks are in order. First, we note that we can rewrite this trace as an integral over the spectrum of $\Sigma_{\text{train}}$:

$$r = \left( 1 - q \int \frac{t}{z - tr} d\mu_{\Sigma_{\text{train}}} (t) \right)^{-1}. \tag{58}$$

It is helpful to compare (58) to the explicit expressions for $g_{44}, g_{53}$ from the linearization before applying the subordination relation, which are

$$g_{44} = 1 + q\bar{\text{tr}} \left[ (z - A)^{-1} A \right] \tag{59}$$

$$g_{53} = \bar{\text{tr}} \left[ (zI - A)^{-1} \Sigma_{\text{train}} \right]. \tag{60}$$

Thus $g_{44} (z)$ is analytic in $z$ everywhere outside the spectrum of $A$, and $g_{44} (\mathbb{H}^\pm) \subset \mathbb{H}^\mp$ and $0 < g_{44} (\mathbb{R}^{<0}) < 1$ (the first inequality is gotten most easily by using $g_{53} < 0$ and $g_{44} = (1 - qg_{53})^{-1}$). In fact, these conditions along with the self-consistent equation (58) are enough to guarantee that the solution is unique, holomorphic, and coincides with $g_{44}$ throughout all of $\mathbb{C}\backslash\mathbb{R}^{\geq 0}$.

**Proposition 4.** *For $z \in \mathbb{C}\backslash\mathbb{R}^{\geq 0}$, there is a unique solution $r(z)$ to (58) satisfying the conditions $r(\mathbb{H}^\pm) \subset \mathbb{H}^\mp$ and $0 < r(\mathbb{R}^{<0}) < 1$. $r(z)$ depends holomorphically on $z$ and can be obtained by iteration of the right hand side of (58) from an arbitrary initial point in $\mathbb{C}\backslash\mathbb{R}^{\geq 0}$.*

*Proof.* Assume $z \in \mathbb{H}^-$. Let $f(r, z)$ be the map defined by the right hand side of (58):

$$f(r, z) := \left( 1 - q \int \frac{t}{z - tr} d\mu_{\Sigma_{\text{train}}} (t) \right)^{-1}.$$

It is straightforward to check that $f(\cdot, z) : \mathbb{H}^+ \to \mathbb{H}^+$. Furthermore, no point on the boundary of $\mathbb{H}^+$ is a fixed point of $f(\cdot, z)$, since $f(\mathbb{R}, z) \subset \mathbb{H}^+$, and $f(\infty, z) = 1$. The Denjoy-Wolff theorem then guarantees that $f(\cdot, z)$ has a unique fixed point in $\mathbb{H}^+$ - and that this point is obtained by iteration of $f(\cdot, z)$ from an arbitrary initial point in $\mathbb{H}^+$. Thus (58) together with the condition $r \in \mathbb{H}^+$ uniquely defines a function $r(z)$ for all $z \in \mathbb{H}^-$.

Now fix $z_0 \in \mathbb{H}^-$. Since $f(\cdot, z_0) : \mathbb{H}^+ \to \mathbb{H}^+$ and $f(\cdot, z_0)$ is not a Möbius transformation (it only can be if $\Sigma_{\text{train}}$ is a scalar matrix), the Schwarz lemma implies $\left|\frac{\partial}{\partial r} f(r(z_0), z_0)\right| < 1$, which means

$$\left|\frac{\partial}{\partial r}(f(r, z_0) - r)\right| = \left|\frac{\partial}{\partial r} f(r, z_0) - 1\right| > 0,$$

and so the implicit function theorem implies there is a holomorphic function solving (58) on some neighborhood of $z_0$ that coincides with $r(z_0)$ at $z_0$. Since $r(z_0) \in \mathbb{H}^+$, this function must also stay in $\mathbb{H}^+$ in some (possibly smaller) neighborhood of $z_0$, and by uniqueness of solutions to (58), this implies that it coincides with $r(z)$ on this neighborhood. Thus $r(z) : \mathbb{H}^- \to \mathbb{H}^+$ is holomorphic at each point of $\mathbb{H}^-$. An identical argument proves the proposition for $z \in \mathbb{H}^+$.

Now suppose $z < 0$. Conjugating the right hand side of (58) by the map $x \mapsto 1/(1 - qx)$ gives a self-consistent equation satisfied by $g_{53}$:

$$y = \int \frac{t}{z - \frac{1}{1-qy}t} d\mu_{\Sigma_{\text{train}}}(t). \tag{61}$$

The condition $0 < r(z) < 1$ implies $g_{53} < 0$. Now letting $h(y, z)$ be the right hand side of (61), $h(y, z) - y$ is convex in $y$ and satisfies $h(0, z) - 0 < 0$ and $h(-\infty, z) - (-\infty) = \infty$, so there is a unique solution to (61) with $y < 0$, and thus a unique solution to (58) with $0 < r(z) < 1$.

Since $\frac{\partial}{\partial y} h(y, z) > 0$, and differentiating $h$ at the fixed point gives

$$\frac{\partial}{\partial y} h(y, z) = \frac{\partial}{\partial y}(1 - qy) \int \frac{t}{z(1-qy) - t} d\mu_{\Sigma_{\text{train}}}(t)$$

$$= 1 - \frac{1}{1 - qy} + z \int \frac{(1-qy)qt}{(z(1-qy) - t)^2} d\mu_{\Sigma_{\text{train}}}(t)$$

$$< 1,$$

$y(z)$ is an attracting fixed point of $h(\cdot, z)$. Since $h(\cdot, z)$ is a conjugate of $f(\cdot, z)$, the unique solution of (58) satisfying $0 < r(z) < 1$ is an attracting fixed point of $f(\cdot, z)$. This implies that there is a neighborhood of $z$ that extends into the upper half plane whose iterates converge to $r(z)$. But since $z < 0$, $f(\mathbb{H}^+, z) \subset \mathbb{H}^+$, and so the Denjoy-Wolff theorem implies that all iterates of $f(\cdot, z)$ initialized in $\mathbb{H}^+$ converge to the same point, which therefore must be $r(z)$.

Finally, $\frac{d}{dy}(h(y) - y) = h'(y) - 1 < 0$ implies that $y(z)$ extends holomorphically to a solution of (61) in an entire neighborhood of $z$. Since $h(y(w), w) - y(w) = 0$ for all $w$ in this neighborhood, at the solution point,

$$y'(z) = -\frac{\frac{\partial}{\partial z} h(y(z), z)}{\frac{\partial}{\partial y}(h(y(z), z) - y(z))} = \frac{\int \frac{t}{\left(\frac{t}{qy-1} + z\right)^2} d\mu_{\Sigma_{\text{train}}}(t)}{\frac{\partial}{\partial y}(h(y(z), z) - y(z))} < 0.$$

A negative derivative implies that for sufficiently small neighborhood $U$ of $z$, $y(U \cap \mathbb{H}^-) \subset \mathbb{H}^+$.

Mapping back to $r(z) := 1/(1 - qy(z))$ yields a holomorphic function satisfying (58) in a neighborhood $U$ of $z < 0$ such that for $r(U \cap \mathbb{H}^-) \subset \mathbb{H}^+$. By uniqueness of solutions in the upper half plane, $r$ must coincide with the function defined earlier on $U \cap \mathbb{H}^-$. Thus $r$ extends holomorphically to the negative real axis. □

The subordination relation implies $g_{44}(z) = r(z)$ in a neighborhood of $\infty$, but both functions extend holomorphically to all of $\mathbb{C}\backslash\mathbb{R}^{\geq 0}$, implying they are equal throughout. This completes the proof of Propositions 2 and 3.

# E  LOCAL GEOMETRY OF MLPS

## E.1  GRADIENT DESCENT DYNAMICS

The initial gradient at $w_0$ is $z := d\nabla\mathcal{L}_{\text{train}}(w_0, \epsilon)$, and the Hessian is $H_{\text{train}} := d\nabla^2\mathcal{L}_{\text{train}}$, so the local approximation for the training loss is

$$\mathcal{L}_{\text{train}}(w) \approx \frac{1}{d} z^\top (w - w_0) + \frac{1}{2d}(w - w_0)^\top H_{\text{train}}(w - w_0),$$

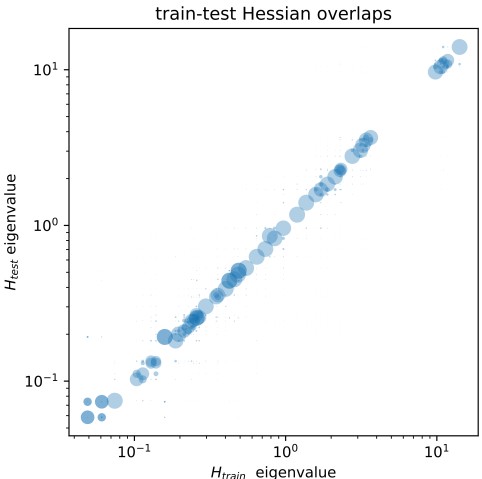

Figure 6: Eigenvector overlap function for one MLP simulation in the context of 3.3. A dot is plotted for every pair of train and test eigenvalues, with dot size and opacity representing squared overlap of the corresponding eigenvectors. Note the very strong train-test alignment indicated by the restriction of almost all overlap to the diagonal.

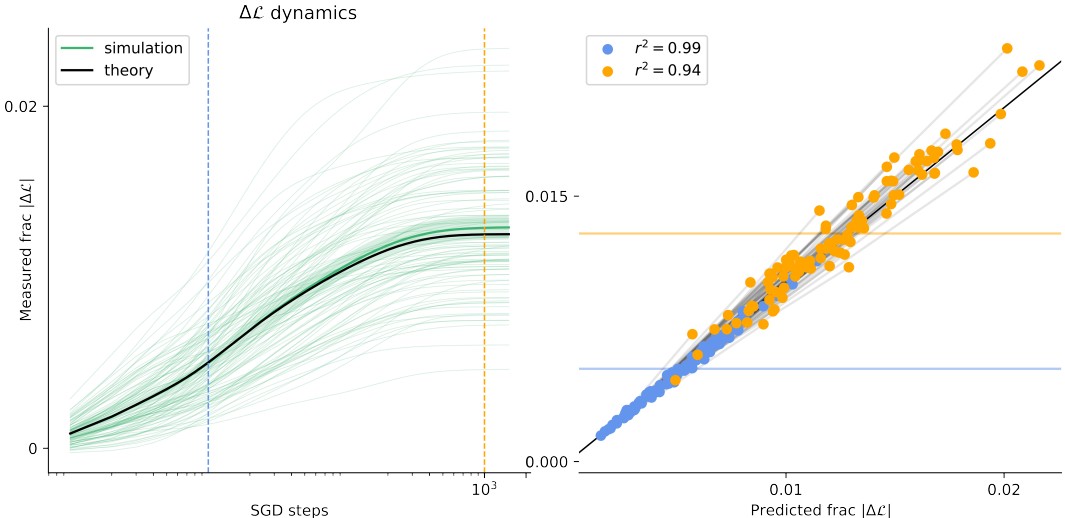

Figure 7: Learning dynamics predicted by local two-loss geometry. Left panel: Test loss trajectories in response to different label noise realizations (individual trajectories shown as thin green lines; average shown as thick green line). Noise amplitude corresponds to red dot in Fig. 4(b). To reduce clutter, only theory average is shown (black line; see (62)). Right panel: measured vs predicted relative test loss increment at two times, corresponding to the vertical blue and gold lines in the left panel. Points corresponding to the same trajectory are connected by gray lines. Horizontal blue and gold lines show means.

where we've discarded additive constants. The gradient is then

$$\nabla\mathcal{L}_{\text{train}}\left(w\right) \approx \frac{1}{d}\left(z - H_{\text{train}}w_0 + H_{\text{train}}w\right),$$

so gradient descent does

$$w \to w - \eta\left(z - H_{\text{train}}w_0 + H_{\text{train}}w\right)$$
$$= \left(I - \eta H_{\text{train}}\right)w - \eta\left(z - H_{\text{train}}w_0\right).$$

Thus,

$$\Delta w_t := w_t - w_0 = \frac{(I - \eta H_{\text{train}})^t - I}{H_{\text{train}}} z,$$

and so the test error satisfies

$$(\Delta \mathcal{L}_{test})_t = \frac{1}{d} g^\top \Delta w_t + \frac{1}{2d} \Delta w_t^\top H_{\text{test}} \Delta w_t \tag{62}$$

$$= \frac{1}{d} g^\top \frac{(I - \eta H_{\text{train}})^t - I}{H_{\text{train}}} z + \frac{1}{2d} z \frac{(I - \eta H_{\text{train}})^t - I}{H_{\text{train}}} H_{\text{test}} \frac{(I - \eta H_{\text{train}})^t - I}{H_{\text{train}}} z, \tag{63}$$

with $g := d\nabla \mathcal{L}_{\text{test}}(w_0)$ and $H_{\text{test}} := d\nabla^2 \mathcal{L}_{\text{test}}(w_0)$.

To compute the GD trajectory for a large network, we precompute

$$z = \text{grad}(\mathcal{L}_{\text{train}})(w_0)$$

$$H_{\text{train}} w_0 = \text{hvp}(\mathcal{L}_{\text{train}}, w_0),$$

set $v_0 := z - H_{\text{train}} w_0$, and simply iterate

$$w \to w - \eta (v_0 + \text{hvp}(\mathcal{L}_{\text{train}}, w)),$$

where grad, hvp compute function gradients and Hessian-vector products.

## F    EFFICIENT CALCULATION OF EIGENVECTOR OVERLAPS FOR LARGE MATRICES

Here we describe the Overlap-KPM algorithm, which estimates the unnormalized or total eigenvector overlaps for two matrices. That is, for self-adjoint $A, B \in \mathbb{R}^{d \times d}$, it estimates the measure

$$\mu_{A,B} := \frac{1}{d^2} \sum_{i,j=1}^d \delta_{\left(\lambda_i^A, \lambda_j^B\right)} \left[ d \left( v_i^A \cdot v_j^B \right)^2 \right],$$

which accumulates all overlap of $\lambda_1$ $A$-eigenspaces with all $\lambda_2$ $B$-eigenspaces. To obtain the normalized overlap function treated in the main text (eg. equation (18)), one simply divides by the spectral densities of $A, B$ at $\lambda_1, \lambda_2$. The problem of estimating spectral densities for large matrices has already received significant attention (see Papyan (2019) in machine learning context), so we assume that one can practically transform back and forth between normalized and unnormalized overlap functions.

### F.1    RANK DEFLATION AND NORMALIZATION

As a preprocessing step, we remove the outlier eigenspaces from each matrix obtained via subspace iteration (cf Fig. 5), and then normalize the spectra to the interval $[-1, 1]$.

### F.2    RANK DEFLATION VIA SUBSPACE ITERATION

The overlap plots of Fig. 5 were generated via Subspace Iteration (SI). SI is a well-known method that generalizes power iteration, so we review it only briefly here. See Papyan (2019) for an explicit implementation in a machine learning context.

Let $M$ be a self-adjoint operator with simple spectrum $\lambda_1, \ldots, \lambda_d$, and take $V \in \mathbb{R}^{d \times k}$ with standard normal entries and then orthonormalize the columns. SI iterates

$$V \to MV$$

$$V \to VQ,$$

where $Q$ is the Gram-Schmidt orthonormalizing upper triangular matrix of $V$. Informally, each application of $M$ amplifies each $i^{th}$ eigenspace coefficient of the columns of $V$ by $\lambda_i$, which generically leads to exponentially greater weight on the leading eigenspaces. The orthonormalization $Q$

prevents all eigenvectors from collapsing onto the same leading eigenvector. Since they are forced to span an $k$-dimensional space, they must converge to the top $k$ eigenvectors of $M$. Overlaps can then be calculated directly by computing pairwise dot products of columns of $V$.

After convergence, outlier eigenspaces are removed from the matrices by replacing each matrix vector product $v \mapsto M(v)$ with

$$v \mapsto M_{def}(v) = M(v) - VV^{\top}v.$$

### F.2.1 Spectrum Normalization

After removing the outlier eigenvalues, one may obtain bounds for the remaining spectrum via standard approaches (eg. the Lanczos algorithm; cf Papyan (2019)). Letting $\lambda_{\min}, \lambda_{\max}$ denote the minimum and maximum eigenvalue (in practice, with a small amount of padding added), we then normalize the matrices to the interval $[-1, 1]$ by replacing $v \mapsto M_{def}(v)$ with

$$v \mapsto M_{norm}(v) = \frac{2}{\lambda_{\max} - \lambda_{\min}} M_{def}(v) - \left(\frac{\lambda_{\max} + \lambda_{\min}}{\lambda_{\max} - \lambda_{\min}}\right) v.$$

### F.3 Overlap-KPM

We now assume the previous preprocessing steps have been performed and in particular that $A, B$'s spectra lie inside $[-1, 1]$.

First note that for kernel function $G$ one can write the kernel-smoothed overlaps exactly as a trace:

$$\bar{\mathrm{tr}}\left[G(A - \lambda_1; \sigma) G(B - \lambda_2; \sigma)\right] = \frac{1}{d^2} \sum_{i,j=1}^{d} G(\lambda_{A,i} - \lambda_1; \sigma) G(\lambda_{B,j} - \lambda_2; \sigma) \left[d(v_{A,i} \cdot v_{B,j})^2\right]. \tag{64}$$

Thus the goal will be to compute such traces for each $(\lambda_1, \lambda_2)$ for some sufficiently small fixed kernel width $\sigma$. Computing such traces directly is prohibitively expensive for very large matrices, and so a standard approach is to use Hutchinson trace estimation, ie. to average $v^{\top}Mv$ over several random samples of, say, standard normal $v$, since

$$\mathbb{E}_v\left[v^{\top}Mv\right] = \mathrm{tr}\left[M\mathbb{E}_v\left[vv^{\top}\right]\right] = \mathrm{tr}\left[M\right].$$

Informal experiments suggested better stability for estimation of PSD traces, so we replace the trace on the left side of (64) with

$$\bar{\mathrm{tr}}\left[G^{1/2}(A - \lambda_1; \sigma) G(B - \lambda_2; \sigma) G^{1/2}(A - \lambda_1; \sigma)\right].$$

Now applying the Hutchinson trick, we sample probes $v_1, \ldots, v_P$ and approximate

$$\bar{\mathrm{tr}}\left[G(A - \lambda_1; \sigma) G(B - \lambda_2; \sigma)\right] \approx \frac{1}{P} \sum_{\mu=1}^{P} v_{\mu}^{\top} G^{1/2}(A - \lambda_1; \sigma) G(B - \lambda_2; \sigma) G^{1/2}(A - \lambda_1; \sigma) v_{\mu}$$

$$= \frac{1}{P} \sum_{\mu=1}^{P} \left\|G^{1/2}(B - \lambda_2; \sigma) G^{1/2}(A - \lambda_1; \sigma) v_{\mu}\right\|^2.$$

To compute the summand, we generalize a standard approach known as the kernel polynomial method. Practically speaking, this entails approximating the kernel functions $G^{1/2}(x - \lambda; \sigma)$ using Chebyshev polynomials $T_j(x)$, which can be computed efficiently using $T_0(x) = 1, T_1(x) = x$, and the recurrence

$$T_j(x) = 2xT_{j-1}(x) - T_{j-2}(x), \qquad j \geq 2.$$

Letting $\alpha, \beta$ be the Chebyshev coefficients of the kernel functions,

$$G^{1/2}(x - \lambda_1; \sigma) = \sum_{i=0}^{\infty} \alpha_i T_i(x)$$

$$G^{1/2}(x - \lambda_2; \sigma) = \sum_{j=0}^{\infty} \beta_j T_j(x),$$

we truncate to degree $K$ and write

$$c_\mu := \left\| G^{1/2} \left( B - \lambda_2; \sigma \right) G^{1/2} \left( A - \lambda_1; \sigma \right) v_\mu \right\|^2 \tag{65}$$

$$\approx \left\| \sum_{i,j=0}^{K} \alpha_i \beta_j T_j \left( B \right) T_i \left( A \right) v_\mu \right\|^2 \tag{66}$$

$$= \sum_{i,j,k,\ell=0}^{K} \alpha_i \beta_j \alpha_k \beta_\ell v_\mu^\top T_i \left( A \right) T_j \left( B \right) T_\ell \left( B \right) T_k \left( A \right) v_\mu \tag{67}$$

$$=: \sum_{i,j,k,\ell=0}^{K} \alpha_i \beta_j \alpha_k \beta_\ell M_{i,j,k,\ell,\mu}. \tag{68}$$

Thus for $P$ probes and order-$K$ Chebyshev truncation, by appropriate choice of the coefficients $\alpha, \beta$, one can approximate a general function from the $P \left( K + 1 \right)^4$ dot products

$$M_{i,j,k,\ell,\mu} = v_\mu^\top T_i \left( A \right) T_j \left( B \right) T_\ell \left( B \right) T_k \left( A \right) v_\mu.$$

This can be improved somewhat using the Chebyshev product identity

$$T_j \left( x \right) T_\ell \left( x \right) = \frac{1}{2} \left( T_{j+\ell} \left( x \right) + T_{|j-\ell|} \left( x \right) \right), \tag{69}$$

so that

$$M_{i,j,k,\ell,\mu} = \frac{1}{2} \left( v_\mu^\top T_i \left( A \right) T_{j+\ell} \left( B \right) T_k \left( A \right) v_\mu + v_\mu^\top T_i \left( A \right) T_{|j-\ell|} \left( B \right) T_k \left( A \right) v_\mu \right),$$

and so all needed dot products can be obtained from the $P \left( K + 1 \right)^2 \left( 2K + 1 \right) \sim 2PK^3$ dot products

$$\tilde{M}_{i,j,k,\mu} := v_\mu^\top T_i \left( A \right) T_j \left( B \right) T_k \left( A \right) v_\mu, \qquad 0 \leq i, k \leq K, \quad 0 \leq j \leq 2K, \qquad 1 \leq \mu \leq P.$$

Algorithm 1 efficiently generates all such probe moments with $\sim PK^2$ matrix vector products. Algorithm 1 actually stores all $2PK^2$ vectors $T_j \left( B \right) T_k \left( A \right) v_\mu$, but in practice, our implementation is significantly more memory efficient. We store all $T_k \left( A \right) v_\mu$, but as $B$'s are added, one only needs to store the current and previous power of $B$. This amounts to $\sim K$ vectors in memory at once.

Once the $\tilde{M}_{i,j,k,\mu}$ are known, equation (68) is used to estimate the trace for each value of $\lambda_1, \lambda_2$, yielding an approximation to the unnormalized overlap function of $A, B$.

Often in machine learning contexts, one or both of $A, B$ has spectrum that is highly peaked around a particular value. For the trace in (64) to accurately reflect the overlaps at $\lambda_1, \lambda_2$, the kernels—more precisely, their finite $K$ Chebyshev series—must decay sufficiently quickly away from $\lambda_1, \lambda_2$ to prevent the spectral spikes from overwhelming the overlap sum. Practically speaking, this can be diagnosed by 1) forming an estimate of $A, B$'s spectral density, eg. using the Lanczos algorithm (see Papyan (2019) for implementation in ML context), 2) forming truncated Chebyshev series for the kernels, and 3) comparing kernel decay to spike height. Insufficient decay usually requires either decreased kernel width $\sigma$, or increased Chebyshev degree $K$ so that polynomial approximations accurately approximate the small tails needed to dampen the spectral spikes.

Algorithm 1 evaluates $O(PK^2)$ matrix vector products. When these correspond to Hessian vector products for a model with $d$ parameters evaluated on $m$ examples, this equates to a total runtime complexity of $O(PK^2md)$. The number of probes $P$ and the Chebyshev degree $K$ are usually small and can be taken to be fixed relative to $m, d$, so runtime is essentially linear in the number of parameters and number of examples. Similarly, Algorithm 1 only requires keeping $O(K)$ matrix vector products in memory at once, for a memory footprint of $O(Kd)$.

Overlap-KPM combines two standard components—Chebyshev polynomial approximation of smooth spectral kernels and Hutchinson trace estimation—and therefore its hyperparameter behavior is straightforward. The truncation order $K$ controls only the polynomial approximation error of the Gaussian kernel; because the kernel is analytic, this error decays exponentially fast in $K$ (Boyd,

1989), and in practice the estimate stabilizes rapidly once $K$ exceeds a modest threshold. The number of probes $P$ affects only the Monte-Carlo variance, which decreases at the usual $O(1/\sqrt{P})$ rate. Empirically, we observe that the estimator is stable over wide ranges of $K, P$ (see tests on synthetic data in F.4).

---

**Algorithm 1:** Overlap-KPM for Eigenvector Overlaps

---

**Input:** $A(v)$, $B(v)$ (normalized MVPs); degree $K$; probes $P$
**Output:** Probe moments $M_{i,j,k,\mu}$ for $0 \leq i, k \leq K$; $0 \leq j \leq 2K$; $1 \leq \mu \leq P$.
**for** $\mu = 1$ **to** $P$ **do**
    sample probe $v_\mu \sim \mathcal{N}(0, I_d)$

    $v_{0,0,\mu} \leftarrow v_\mu; \quad v_{0,1,\mu} \leftarrow A(v_\mu)$
    **for** $i = 2$ **to** $K$ **do**
        $v_{0,i,\mu} \leftarrow 2A(v_{0,i-1,\mu}) - v_{0,i-2,\mu}$

    **for** $k = 0$ **to** $K$ **do**
        $v_{1,k,\mu} \leftarrow B(v_{0,k,\mu})$
        **for** $j = 2$ **to** $2K$ **do**
            $v_{j,k,\mu} \leftarrow 2B(v_{j-1,k,\mu}) - v_{j-2,k,\mu}$

    **for** $i = 0$ **to** $K$ **do**
        **for** $k = 0$ **to** $K$ **do**
            **for** $j = 0$ **to** $2K$ **do**
                $M_{i,j,k,\mu} \leftarrow v_{0,i,\mu} \cdot v_{j,k,\mu}$

---

### F.4 TESTS ON SYNTHETIC DATA

Algorithm 1 with gaussian kernel is applied to synthetic data in Fig. 8. $A, B \in \mathbb{R}^{1000 \times 1000}$ are generated according to

$$A = W_1, \qquad B = W_2 + A^2, \tag{70}$$

where $W_1, W_2$ are independent Wishart matrices with aspect ratio $\alpha = 5$. The left panel shows the ground truth gaussian-smoothed overlap function of $A, B$. Note the nontrivial alignment due to $B$'s dependence on $A$. The right panel shows the approximation generated via Overlap-KPM, showing good qualitative match.

We performed informal experiments varying $K, P$ to test the robustness of Algorithm 1 (Figure 9). As expected, accuracy quickly improves and eventually saturates as the Chebyshev approximation order $K$ is increased. The variance of the estimator as a function of $P$ decays as $O(1/\sqrt{P})$ with a constant of proportionality that depends on the input matrices. As figure 9 shows, even for a modest number of probes (eg., $P = 4$), results can be quite accurate.

### F.5 HESSIAN OVERLAPS OF RESNET-20

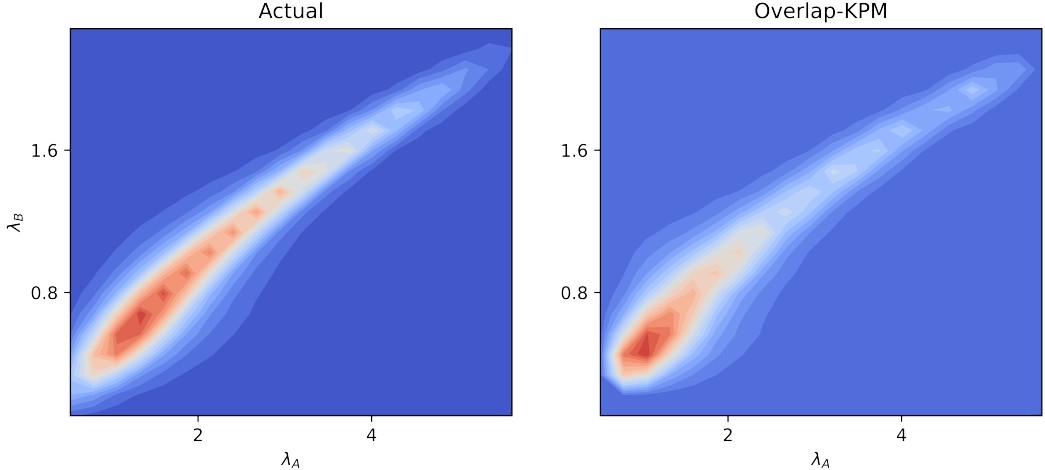

Figure 8: Overlap-KPM algorithm recovers overlaps on synthetic data. $A = Z_1^\top Z_1/m$ and $B = Z_2^\top Z_2/m + A^2$, where $Z_1, Z_2 \in \mathbb{R}^{m,d}$ are independent matrices with iid. standard normal entries. $d = 1000, \alpha := m/d = 5$. Chebyshev degree: $K = 45$; number of probes $P = 4$. Left panel shows actual eigenvector overlaps at eigenvalues $\lambda_A, \lambda_B$, smoothed with a gaussian kernel of width $1/16$. Right panel shows approximation derived from the Overlap-KPM algorithm.

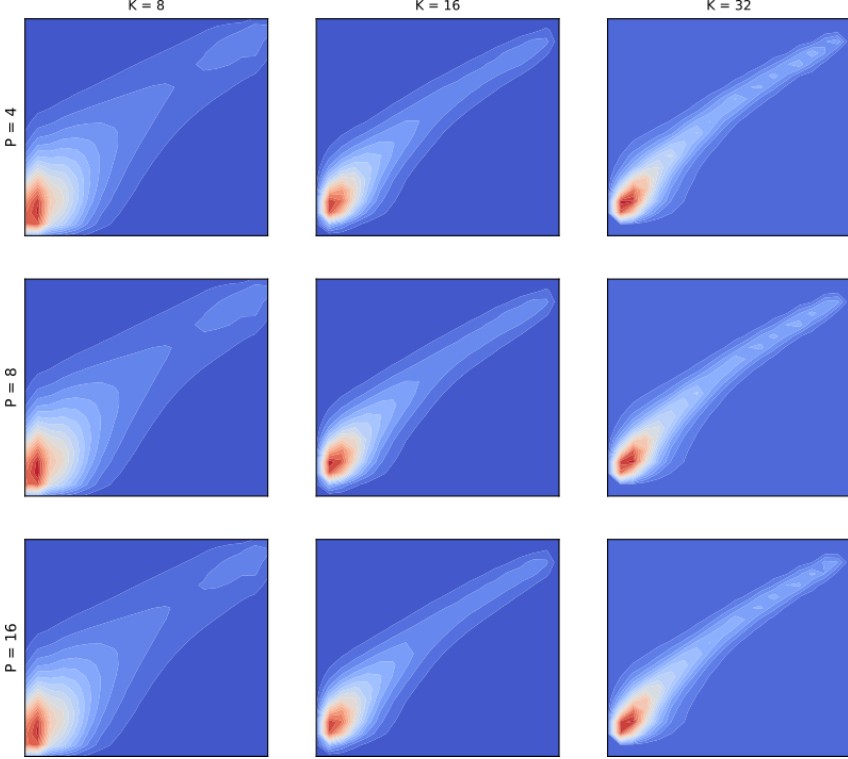

Figure 9: Varying $K, P$ in overlap-KPM. Gaussian kernel with width of $1/32$. Matrices $A, B$ were generated as in 8

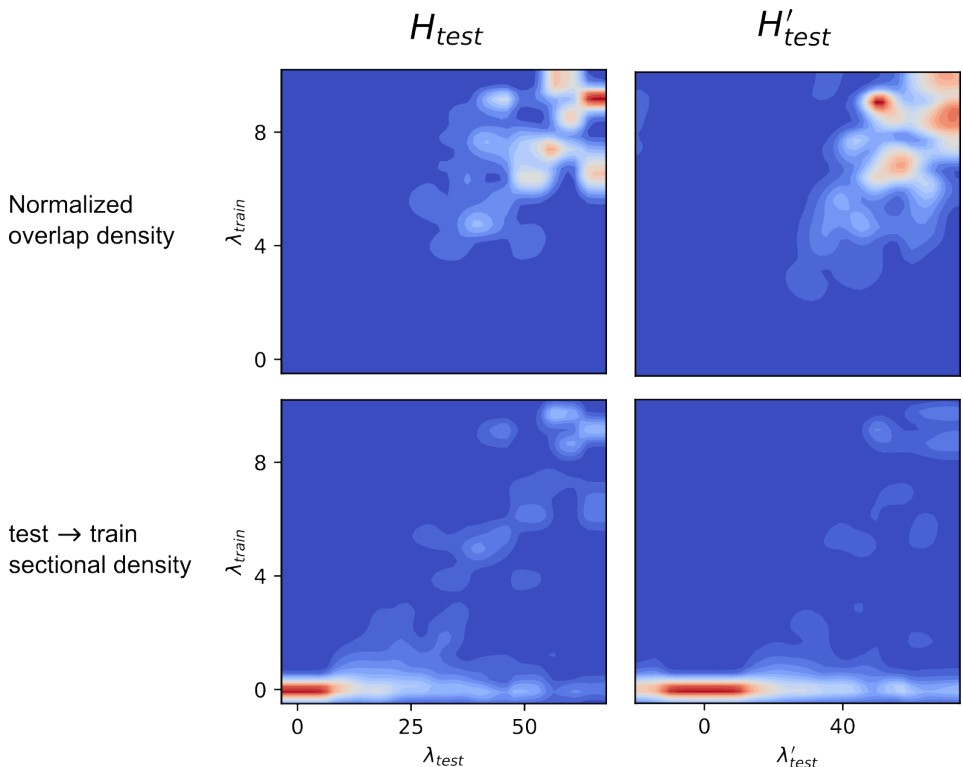

Figure 10: Overlap-KPM algorithm for Hessian overlaps of CIFAR-10-trained ResNet-20. Degree $K = 45$ and $P = 4$ probes with Jackson smoothing applied to gaussian kernel Chebyshev coefficients. Left column shows overlaps for $H_{\text{train}}, H_{\text{test}}$ (ie. balanced test set), while right column shows overlaps for $H_{\text{train}}, H'_{\text{test}}$ (imbalanced test set). Top row shows normalized overlap functions $O_{H_{\text{train}}, H_{\text{test}}}(\lambda_{\text{train}}, \lambda_{\text{test}})$ and $O_{H_{\text{train}}, H'_{\text{test}}}(\lambda_{\text{train}}, \lambda'_{\text{test}})$. For ease of visualization, bottom row shows test, train sectional densities $O_{H_{\text{train}}, H_{\text{test}}}(\lambda_{\text{train}}, \lambda_{\text{test}})\mu_{\text{train}}(\lambda_{\text{train}})$ and $O_{H_{\text{train}}, H'_{\text{test}}}(\lambda_{\text{train}}, \lambda'_{\text{test}})\mu_{\text{train}}(\lambda_{\text{train}})$—the average overlap of 1-D $H_{\text{test}}/H'_{\text{test}}$ eigenspaces onto *full* eigenspaces of $H_{\text{train}}$. In both rows, strong diagonal overlaps are visible in the left column that are reduced or absent in the right column. Note also in the bottom row that the tail of the $\lambda_{\text{train}} \approx 0$ band extends significantly further for $H'_{\text{test}}$ than for $H_{\text{test}}$, indicating significant loss of high $H'_{\text{test}}$ eigenspace energy into the low-eigenvalue band of $H_{\text{train}}$.

