# OpenReview forum: "Beyond Spectra: Eigenvector Overlaps in Loss Geometry"
_ICLR.cc/2026/Conference — ICLR 2026 Poster_

### Official Review · Reviewer_tXsV · 2025-10-27

**Soundness:** 3
**Presentation:** 2
**Contribution:** 3
**Rating:** 8
**Confidence:** 3

**Summary:**

The authors first formalize the local fluctuation at a local minimum of a model, and then the authors analyze the eigen space of the Hessian of the training loss landscape during these fluctuations. Finally, the authors relate the training Hessian to the Hessian eigen space of the test loss landscape to explain loss behavior during covariate shift and commonly studied optimization behavior, such as multiple descent.

**Strengths:**

1. The reviewer believes that the contribution of this paper is strong. Through analyzing the loss landscape between training and testing, the authors explain the behavior of a couple of important scenarios in modern deep learning, such as covariate shifts (commonly happen in applications), and multiple descent (commonly observed during optimization of over-parameterized models).
2. The author not only provides analysis and empirical results with synthetic training/testing environments but also extends their findings to more complex datasets such as CIFAR.
3. The work is novel.

**Weaknesses:**

Minor:
1. The related work seems to slightly lag behind, for exammple, regularizers that encourage cross-domain invariance have recent advancements such as [1]. Maybe the related work can also touch on sharpness-aware optimization and other robust optimization techniques that focus on controlling the gradient and the Hessian. It will round out the related work nicely to tie it back to the more application side of machine learning.
2. A suggestion on notation: in eq 6, a notation $q$ is used, but it was only defined until the page after. Then also a parameter $\alpha = q^{-1}$ is used through out the paper. Perhaps the author can tidy up this notation and stick with $\alpha$

Major:
1. The presentation of the work can be improved. The reviewer sincerely hope that the authors can use the additional page during rebuttal to strengthen/clarify some of the theoretical points and figures in the manuscript. See following for details.


[1]: Hasan, Ali, Haoming Yang, Yuting Ng, and Vahid Tarokh. "Elliptic Loss Regularization." In The Thirteenth International Conference on Learning Representations.

**Questions:**

1. Perhaps the progression from eq 1 to eq 2 can be improved with a slight introduction on how eq 2 is achieved? Similarly, this can be improved through the development of eq 5-7 (or refer to the appendix).
2. On line 156, the authors assumed $\mathbb{E}z = 0$, is this assumption realistic during scenarios such as covariate shift? If the noise $\epsilon$ is caused by covariate shift while the training data is centered, does this mean $\mathbb{E}z \neq 0$?
3. How should one intuitively understand the paragraph from lines 214 to 217. Does this essentially mean that if the noisy part of the train landscape aligns with the more important part of the test landscape, we will see increased loss value during evaluation?
4. In Proposition 1, can the authors explain conceptually what *"X is free from A, B"* means? Does it simply mean independence?
5. Figure 1 a) seems confusing. What is the purpose of the red contour in Figure 1a)? How are they related to the cyan? Figure 1 b) and c) are cleaner and more understandable.
6. Figure 3 b), does each color of the contour mean a specific ranking of eigenvalue?

---

> ### Author Response · Authors · 2025-11-20
> **Response 1 to Reviewer tXsV [1/2]**
>
> **Weaknesses**
>
> *Minor*
>
> *The related work seems to slightly lag behind, for exammple, regularizers that encourage cross-domain invariance have recent advancements such as [1]. Maybe the related work can also touch on sharpness-aware optimization and other robust optimization techniques that focus on controlling the gradient and the Hessian. It will round out the related work nicely to tie it back to the more application side of machine learning.*
>
> Thank you for these suggestions. As also mentioned in our response to reviewers Rqbg and G7EJ, we have substantially expanded the related work to discuss prior work on eigenvector overlaps in ML (Wu et al., 2022; May et al., 2019), domain-invariance methods (including Elliptic loss regularization), and sharpness-aware optimization. These additions place our contributions in clearer relation to existing approaches.
> &nbsp;
> &nbsp;
> *A suggestion on notation: in eq 6, a notation $q$ is used, but it was only defined until the page after. Then also a parameter $\alpha = q^{-1}$ is used through out the paper. Perhaps the author can tidy up this notation and stick with $\alpha$.*
> &nbsp;
> Agreed. We now write all formulas in terms of $\alpha$.
> &nbsp;
> &nbsp;
> *Major*:
>
> *The presentation of the work can be improved. The reviewer sincerely hope that the authors can use the additional page during rebuttal to strengthen/clarify some of the theoretical points and figures in the manuscript. See following for details.*
>
> Thank you for your suggestions. As noted in our response to several other reviewers, we have substantially reorganized the theory section in order to make the presentation clearer:
> 1. We have included a new preliminaries section explaining notation and the approximation scheme used to derive the fluctuation law (beginning of section 3.1, 3.1.1).
> 2. We have encapsulated the theoretical results of section 3 into two new theorems to improve the organization.
> 3. Several small issues and abuses of notation have been corrected.

---

> ### Author Response · Authors · 2025-11-20
> **Response 2 to Reviewer tXsV [2/2]**
>
> *Questions*:
>
> 1. *Perhaps the progression from eq 1 to eq 2 can be improved with a slight introduction on how eq 2 is achieved? Similarly, this can be improved through the development of eq 5-7 (or refer to the appendix).*
> &nbsp;
> The progression from equations (1)-(3) has been significantly reworked. Equation (3) now appears as Theorem 1. A proof sketch follows immediately below and a detailed derivation appears in Appendix B.2.
> &nbsp;
> 2. *On line 156, the authors assumed $\mathbb{E}z = 0$, is this assumption realistic during scenarios such as covariate shift? If the noise $\epsilon$ is caused by covariate shift while the training data is centered, does this mean $\mathbb{E}z \neq 0$?*
> &nbsp;
> We apologize for the confusion. In our notation, $z$ is the first-order perturbation of the training loss induced by the small parameter $\epsilon$. The assumption $\mathbb{E}z=0$ concerns perturbations of the training loss, not covariate shift. Covariate shift, in contrast, changes the test loss while leaving the training loss (and hence $z$) unchanged. Thus, under covariate shift one still has $\mathbb{E}z = 0$; what changes is the test Hessian, not the perturbation of the training loss. We have edited the text of 3.1.1 to make this issue clearer.
> &nbsp;
> 3. *How should one intuitively understand the paragraph from lines 214 to 217. Does this essentially mean that if the noisy part of the train landscape aligns with the more important part of the test landscape, we will see increased loss value during evaluation?*
> &nbsp;
> Yes—perturbations to the training loss induce displacements in the learned parameters. When these displacements align strongly with directions of large test loss sensitivity, error is amplified. The order of this paragraph in the text has been adjusted.
> &nbsp;
> 4. *In Proposition 1, can the authors explain conceptually what "X is free from A, B" means? Does it simply mean independence?*
> &nbsp;
> Freeness is a notion of independence that is suited to large random matrices and holds asymptotically for a wide range of common random matrix models, including, eg. matrices with independent, mean-0 entries. In the manuscript, this result is only applied to large gaussian matrices, for which freeness holds. We’ve added a sentence to the text after the statement of the transfer law clarifying this point.
> &nbsp;
> 5. *Figure 1 a) seems confusing. What is the purpose of the red contour in Figure 1a)? How are they related to the cyan? Figure 1 b) and c) are cleaner and more understandable.*
> &nbsp;
> Thank you for pointing this out. In Fig. 1a, the cyan and red contours represent the train and test Hessians (equivalently, the quadratic parts of the train and test losses). The purpose of the figure is to show that under covariate shift, the two quadratic forms need not share eigendirections, and that this misalignment—rather than spectral differences alone—drives the change in test loss. We have revised the caption to make this clearer.
> &nbsp;
> 6. *Figure 3 b), does each color of the contour mean a specific ranking of eigenvalue?*
> &nbsp;
> Each horizontal slice of Figure 3(b) represents the spectral density of $H_{\text{train}}$ at the corresponding value of $\alpha$. For ease of visualization, each horizontal slice is normalized to a maximum value of 1, and the resulting values are quantized. We realize that the quantization may have caused confusion, so it is now explicitly mentioned in the caption.

---

> > ### Comment · Reviewer_tXsV · 2025-11-22
> >
> > The reviewer has addressed many of my previous concerns, especially with the improved structure of the paper and the expanded discussion with other related work. I would like to maintain my score

---

### Official Review · Reviewer_GPhb · 2025-10-29

**Soundness:** 4
**Presentation:** 3
**Contribution:** 4
**Rating:** 8
**Confidence:** 4

**Summary:**

The authors derive a universal local fluctuation law considering the train/test loss Hessian spectra and overlaps of their eigenspaces, and apply it to ridge regression, trying resolve the puzzle of multiple descent.
They also empirically validate their theory with MLPs.

**Strengths:**

- This paper provides a new perspective on the importance of the eigenspace overlaps.
- It explains the multiple descent phenomenon from this perspective.

**Weaknesses:**

- As (3) is very important equation in the paper, I'd like to know how to obtain the equation in detail. The explanation is very unclear to me.
    - The approximations of (1) and (2) are not clear. In what sense, they are approximated? (e.g., $O(\\|w-w_0\\|^2), O(\\|\epsilon\\|^2)$)
    - What is the definition of
        - $J_{\text{train}}(w,\epsilon)$ (Is it $J\_{\text{train}}(w+\epsilon)$?)
        - $H_{\text{train}}$? If it depends on $w$ as written in the paper ($H_{\text{train}}:=d\nabla^2 J_{\text{train}}(w,0)$), then (1) is not quadratic and $\Delta w$ is not just $-H^{-1}_{\text{train}}z$.
        - $\Delta J_{\text{test}}$ (Is it $J_{\text{test}}(w_0+\Delta w,\epsilon)-J_{\text{test}}(w_0,\epsilon)$? or without $\epsilon$?)
    - The approximations of (1) and (2), but equality in (3). How do we get the exact equality in (3) from (1) and (2)?
- The caption of Fig 2a says "$J_\text{{\color{red}train}},\Delta J_{\text{test}}$ and bias" but in the panel it says "$J_{\text{test}},\Delta J_{\text{test}}$, Bias."
- em dash? (L41, L222, L262, L348, L353, L361, ...)
- errata? $\mu_\Sigma = p_1\delta_{s_1^{\color{red}2}}+p_2\delta_{s_2^2}$
- What is $\tilde\lambda{\color{red}'}$ in (9)?
- errata? (L265) Isn't it $s_1^2,s_2^2=2^0,2^{-4}$? The eigenvalues in Fig 1(b) should be $s_i^2$ as written in (8).

**Questions:**

see weaknesses

---

> ### Author Response · Authors · 2025-11-20
> **Response to Reviewer GPhb**
>
> We thank reviewer GPhb for several helpful suggestions. We have substantially reorganized the theory section in order to make the presentation clearer.
> 1. We have included a new preliminaries section explaining notation and the approximation scheme used to derive the fluctuation law (beginning of section 3.1).
> 2. We have encapsulated the theoretical results of section 3 into two new theorems to improve the organization.
> 3. Several small issues and abuses of notation have been corrected.
> &nbsp;
>
> *Detailed comments*
>
> 1. *As (3) is very important equation in the paper, I'd like to know how to obtain the equation in detail. The explanation is very unclear to me.*
> &nbsp;
> The progression from equations (1)-(3) has been significantly reworked. Equation (3) now appears as Theorem 1. A proof sketch follows immediately below and a detailed derivation appears in Appendix B.2.
> &nbsp;
> 2. *The approximations of (1) and (2) are not clear. In what sense, they are approximated? (e.g., $O(\Vert w-w_0\Vert^2), O(\Vert\epsilon\Vert^2)$)*
> &nbsp;
> The exact approximation scheme is now clarified in section 3.1, 3.1.1.
> &nbsp;
> 3. *What is the definition of
> $J_{\text{train}}(w,\epsilon)$ (Is it $J_{\text{train}}(w+\epsilon)$?)*
> &nbsp;
> We do not assume that the perturbation $\epsilon$ enters the loss additively; the $\epsilon$-dependence of J_{\text{train}}​ is left completely generic. The only requirement for our analysis is that the perturbation induces small parameter displacements for which a quadratic approximation remains valid. This can be ensured under mild regularity conditions—e.g., smoothness or differentiability of the loss with respect to $\epsilon$, depending on the level of guarantee needed. The text has been adjusted to make this clearer.
> &nbsp;
> 4. *$H_{\text{train}}$? If it depends on $w$ as written in the paper ($H_{\text{train}}:=d\nabla^2 J_{\text{train}}(w,0)$), then (1) is not quadratic and $\Delta w$ is not just $-H^{-1}_{\text{train}}z$.*
> &nbsp;
> In our revised manuscript, we chose to present the derivation using quadratic surrogate losses because this gives the simplest and cleanest exposition of the fluctuation formulas. However, we note that one can obtain a formally identical theory that is asymptotically exact using a path-averaged train Hessian. This yields an effective train Hessian that makes the displacement equation $\Delta w=-H_{\text{eff}}^{-1}z$ exact, while the test loss continues to use its ordinary Hessian, so the leading-order test-loss increment retains the same quadratic form used in the main text. In this formulation, one recovers the same decomposition of the test-loss increment in terms of eigenvalues and the overlaps of the test Hessian onto the (effective) train-Hessian eigenspaces. We have added a short appendix outlining this construction (Appendix B.2.1).
> &nbsp;
> 5. *$\Delta J_{\text{test}}$ (Is it $J_{\text{test}}(w_0+\Delta w,\epsilon)-J_{\text{test}}(w_0,\epsilon)$? or without $\epsilon$?)*
> &nbsp;
> Note that in our treatment $\epsilon$ is a perturbation that applies to the train loss only. The fluctuation law quantifies the effect of this perturbation on a fixed test loss. This point is clarified and $\Delta J_{\text{test}}$ (now $\Delta \mathcal{L}$) is now defined explicitly in the notation section.
> &nbsp;
> 6. *The approximations of (1) and (2), but equality in (3). How do we get the exact equality in (3) from (1) and (2)?*
> &nbsp;
> Fixed. See previous responses.
> &nbsp;
> 7. *The caption of Fig 2a says "$J_\text{{\color{red}train}},\Delta J_{\text{test}}$ and bias" but in the panel it says "$J_{\text{test}},\Delta J_{\text{test}}$, Bias."*
> &nbsp;
> Thank you for bringing this to our attention. The notation has been updated and the offending line has been fixed.
> &nbsp;
> 8. *em dash? (L41, L222, L262, L348, L353, L361, ...)*
> 9. *errata? $\mu_\Sigma = p_1\delta_{s_1^{\color{red}2}}+p_2\delta_{s_2^2}$*
> &nbsp;
> Fixed.
> &nbsp;
> 10. *What is $\tilde\lambda{\color{red}'}$ in (9)?*
> &nbsp;
> The definition of $\tilde{\lambda}$ now appears in the main text as part of Theorem 3. We've replaced the notation $\tilde{\lambda}'$ with $d\tilde{\lambda}/d\lambda$.
> &nbsp;
> 11. *errata? (L265) Isn't it $s_1^2,s_2^2=2^0,2^{-4}$? The eigenvalues in Fig 1(b) should be $s_i^2$ as written in (8).*
> &nbsp;
> Thank you. Done.

---

> > ### Comment · Reviewer_GPhb · 2025-11-26
> >
> > Thank you for the clarifications (e.g., "ε representing a general training perturbation. We remain agnostic about the source of
> > the perturbation"). I found that the manuscript has become much clearer. I still support accepting this paper.

---

### Official Review · Reviewer_zuhF · 2025-10-29

**Soundness:** 2
**Presentation:** 2
**Contribution:** 3
**Rating:** 6
**Confidence:** 2

**Summary:**

This paper studies the use of loss geometry for both train and test sets, specifically the eigenvector overlap between the two, rather than just their spectra, as measure of loss geometry and to predict generalization. It shows that this measure explains covariate shift and multiple descent through a unified lens via simulations on synthetic data. It also develops an efficient estimator to show how this measure can quantify how class imbalance can induce misalignment in train-test loss geometry.

**Strengths:**

The paper makes interesting contributions to use eigenvalue overlap between train and test Hessians, rather than just their spectra, as measure of loss geometry and generalization, and presents a scalable numerical estimator and empirical validations.

**Weaknesses:**

The paper is missing an expanded related work section with a detailed discussion on most closely related works. As someone who is not very familiar with the literature on random matrix theory, and eigenspace overlap estimators, it is hard to judge the novelty of this work relative to other works. It would be good to add a more detailed discussion comparing and contrasting the work with most closely related work.

The experimental methodology in Section 3.4 is somewhat strange. It is stated that “a CIFAR-10 trained ResNet-20 was obtained from Chen”, and then “5000 train and test examples were randomly selected to define train and test Hessians”. A more convincing experiment would be to train it from scratch using the selected train set samples. Additionally, why not use imbalanced train set with balanced test set to compare the effect of imbalance?

**Questions:**

Can authors add details on compute and runtime for the CIFAR-10 results in Section 3.4? Is it possible to use the proposed method for larger models?

Suggestions to improve writing/readability:

1. The paper uses hyphens instead of em dashes in almost every occurrence, and in some cases, it uses en dashes instead of hyphens, please fix.
2. The paper uses $J$ to denote the loss. I suggest using $\mathcal{L}$, or simply $L$, which is more standard. $J$ could be misinterpreted as denoting the Jacobian.
3. In line 157, the expectation terms are missing square brackets.
4. In line 162, the order of $H_{\text{test}}$ and $C_{\text{train}}$ is swapped.
5. In line 215, it should be ‘significant’.
6. In Eq. (8), it should be $\delta_{s_2}^2$.
7. In Fig. 1, panel (b) is missing the y-axis label, please clarify.
8. In most cases, the word ‘traces’ is used to refer to the solid lines in the plots. I suggest simply using ‘solid lines’.
9. In most cases the subfigures are referred to as, e.g., Fig. 1a instead of Fig. 1(a), please fix.
10. Fig. 2 caption states “Traces in panel a) correspond to gold and blue lines”. Please clarify.
11. In line 323, the phrase “geometric cartoon” should be rephrased.
12. In line 377, ‘2d’ should be ‘2D’.

---

> ### Author Response · Authors · 2025-11-20
> **Response 1 to Reviewer zuhF [1/2]**
>
> **Weaknesses**
>
> *The paper is missing an expanded related work section with a detailed discussion on most closely related works. As someone who is not very familiar with the literature on random matrix theory, and eigenspace overlap estimators, it is hard to judge the novelty of this work relative to other works. It would be good to add a more detailed discussion comparing and contrasting the work with most closely related work.*
>
> We have substantially expanded the introduction to situate our contribution within several relevant lines of work. First, we now discuss two machine learning studies that explicitly analyze eigenspace overlap (Wu et al., 2022; May et al., 2019). While their focus differs from ours, they highlight the relevance of directional information beyond spectra. Second, we added a fuller account of work on domain invariance, which seeks robustness across multiple domains through regularizers that align gradients, features, or Hessians. Third, we included a discussion of sharpness-aware and curvature-based optimization, which—like our analysis—connect geometry to generalization, but do so in a single-loss setting. In contrast, our framework is explicitly two-loss, targeting geometric effects that arise only when comparing the train and test losses.
> &nbsp;
> &nbsp;
> *The experimental methodology in Section 3.4 is somewhat strange. It is stated that “a CIFAR-10 trained ResNet-20 was obtained from Chen”, and then “5000 train and test examples were randomly selected to define train and test Hessians”. A more convincing experiment would be to train it from scratch using the selected train set samples. Additionally, why not use imbalanced train set with balanced test set to compare the effect of imbalance?*
>
> Thank you for these suggestions. We realize that the motivation for the setup in Section 3.4 was not sufficiently explicit. The goal of this experiment is not to test the quantitative accuracy of the fluctuation formula. Instead, it is designed to:
> 1. Demonstrate the practical use of our Hessian-overlap estimators in a modern neural network (a CIFAR-10 ResNet-20), and
> 2. Show that a common type of domain change—class imbalance in the test set—indeed produces a measurable change in two-loss geometry, as captured by the train-test Hessian eigenvector overlaps.
>
> For this purpose, the most direct and controlled setup is to:
> 1. Fix a single trained model and its training loss geometry,
> 2. Estimate its train Hessian using a manageable subset of 5000 training examples, and
> 3. Define two different test losses: one using a balanced class distribution and one restricted to a subset of classes, and then compare their “two-loss” geometric structure.
>
> This design isolates exactly the phenomenon the formulas are meant to clarify—the change in two-loss geometry induced by a controlled modification of the test distribution—while keeping the underlying trained model fixed. Training a new model on the 5000-sample subset, while interesting, would change the learned features and Hessian spectrum and thus test a different question. Similarly, while one could construct many other imbalance scenarios, the balanced-vs-subset contrast already provides a clean and illustrative example of how domain shift shows up in the Hessian-overlap geometry.
>
> We have revised the text to make this experimental rationale and scope clear. While time constraints during the discussion period limit our ability to perform substantially larger-scale experiments, we would welcome further reviewer guidance on which specific extensions or imbalance scenarios would be most valuable to explore in this or future work.
> &nbsp;
> &nbsp;
> **Questions**
>
> *Can authors add details on compute and runtime for the CIFAR-10 results in Section 3.4? Is it possible to use the proposed method for larger models?*
>
> The CIFAR-10 experiment was run using standard PyTorch Hessian-vector products on a personal machine, with a total runtime of a few hours.
>
> In terms of scaling: in our notation, P denotes the number of Hutchinson probe vectors and K the degree of the Chebyshev approximation. In practice, both P and K are small and may be regarded as fixed constants relative to the model dimension. Under this regime, the estimator requires $O(P K^3)$ Hessian-vector products in total, while storing only $O(K)$ vectors at any time. The cost of a single HVP is equivalent to one backward pass and therefore scales linearly with model size (and with the number of data points used to estimate the Hessians). As a result, the overall runtime and memory footprint scale essentially linearly in model dimension, making the method practical for substantially larger architectures.
>
> These points are now clarified in a short remark in the main text and additional text in Appendix F.3.

---

> > ### Comment · Reviewer_zuhF · 2025-11-26
> >
> > Thank you for your response, and updates to the paper to incorporate some of the suggestions.
> >
> > I am still not convinced by the response related to the CIFAR-10 experiment in Section 3.4. Originally, I had two concerns:
> >
> > 1. Using the same set of samples to train the model and compute the train Hessian (to ensure better estimation), and
> > 2. Validating if the Hessian-overlap estimator is useful in a setting where the train set is imbalanced while test is balanced (to consider another setting where the overlap estimator can be practically useful).
> >
> > While your response for point 2, that "while one could construct many other imbalance scenarios, the balanced-vs-subset contrast already provides a clean and illustrative example of how domain shift shows up in the Hessian-overlap geometry", is not ideal, it can be considered acceptable: considering different imbalance scenarios could be left to future work, as long as this work presents one concrete scenario demonstrating the utility of the method.
> >
> > Now, for point 1, the main issue is that in the current approach, the train Hessian is not estimated exactly. I agree about points 1 and 3 as requirements for a controlled setup, but not about point 2. It should at least be supported with some evidence, e.g., using some other subset of 5000 samples does not change the train Hessian estimation by a lot. Without that, it is not convincing that the method would be predictive in other settings with imbalance. Similarly, how does the test distribution, e.g., extent of imbalance, impact the results? Can the overlap estimator capture shifts that are less extreme than the current one, in this setting? I suggest adding these experiments to strengthen the paper.
> >
> > Other than this, the rebuttal has addressed my other concerns, and I will maintain my score for now.

---

> ### Author Response · Authors · 2025-11-20
> **Response 2 to Reviewer zuhF [2/2]**
>
> *Suggestions to improve writing/readability:*
>
> 1. *The paper uses hyphens instead of em dashes in almost every occurrence, and in some cases, it uses en dashes instead of hyphens, please fix.*
> &nbsp;
> Done. Thank you for pointing this out.
> &nbsp;
> 2. *The paper uses $J$ to denote the loss. I suggest using $\mathcal{L}$, or simply L, which is more standard. $J$ could be misinterpreted as denoting the Jacobian.*
> &nbsp;
> Thank you for the suggestion. We’ve changed our notation for the loss from $J$ to $\mathcal{L}$.
> &nbsp;
> 3. *In line 157, the expectation terms are missing square brackets.*
> 4. *In line 162, the order of $H_{\mathrm{test}}$ and $C_{\mathrm{train}}$ is swapped.*
> 5. *In line 215, it should be ‘significant’.*
> &nbsp;
> Done.
> &nbsp;
> 6. *In Eq. (8), it should be $\delta^2_{s_2}$.*
> &nbsp;
> Fixed. Eq. (8) now reads $\mu_{\Sigma}:=p_1\delta_{s_1^2}+p_2\delta_{s_2^2}$.
> &nbsp;
> 8. *In Fig. 1, panel (b) is missing the y-axis label, please clarify.*
> 9. *In most cases, the word ‘traces’ is used to refer to the solid lines in the plots. I suggest simply using ‘solid lines’.*
> 10. *In most cases the subfigures are referred to as, e.g., Fig. 1a instead of Fig. 1(a), please fix.*
> &nbsp;
> Done.
> &nbsp;
> 11. *Fig. 2 caption states “Traces in panel a) correspond to gold and blue lines”. Please clarify.*
> &nbsp;
> We’ve updated the wording of Fig. 2’s caption to clarify that the gold and blue lines in (a) and (b) correspond to one another.
> &nbsp;
> 12. *In line 323, the phrase “geometric cartoon” should be rephrased.*
> 13. *In line 377, ‘2d’ should be ‘2D’.*
> &nbsp;
> Done.
> &nbsp;

---

### Official Review · Reviewer_G7EJ · 2025-11-03

**Soundness:** 3
**Presentation:** 2
**Contribution:** 3
**Rating:** 2
**Confidence:** 3

**Summary:**

This paper studies how the relationship between the 2nd order shape of the train and test objectives affects generalization. They then use the general results to study several regression settings, including covariate shift. Finally, they extend the results to neural networks.

**Strengths:**

The results are interesting and provide an additional perspective from which to view generalization. I am not familiar enough with the details of the related literature to know if this is the first work to consider this, but I trust other reviewers will know more. The paper also contains a very large number of results.

**Weaknesses:**

As written, this paper reads like a physics paper. In order to be published in a computer science / ML venue, I think it needs a bit more exposition describing notation, exactly the approximations being made in Section 3, and what the operators in 3.1.2 represent in the ML setting.

1. Is $\Delta J_\text{test} = J_\text{test}(w_0 + \Delta w) - J_\text{test}(w_0)?$
2. What is the source of the noise $\epsilon$? Is it sampling noise, label noise, general?
3. It would help to break up Section 3 into Theorem statements.
4. It would help to have prose at the start of Section 3, 3.1 describing what those sections do.
5. Consider separating out preliminaries for notation and results. The authors should be more explicit about things like what the noise is, what exactly $\Delta J_\text{test}$ is, etc.

It seems like a nice paper, but I would recommend either submitting it to a different venue or writing it more in the style of ML theory papers for publication at an ML conference.

**Questions:**

Feel free to clarify the points above.

---

> ### Author Response · Authors · 2025-11-20
> **Response to Reviewer G7EJ**
>
> **Weaknesses**
>
> *As written, this paper reads like a physics paper. In order to be published in a computer science / ML venue, I think it needs a bit more exposition describing notation, exactly the approximations being made in Section 3, and what the operators in 3.1.2 represent in the ML setting.*
>
> Thank you for these suggestions. We agree that the original draft needed clearer exposition and better ML-oriented framing, especially in Section 3. We have substantially revised this part of the paper as follows:
> - Notation and preliminaries have been separated into a dedicated subsection at the start of Section 3.1, following standard ML-theory conventions.
> - The approximation scheme used to derive the fluctuation law is now stated explicitly in 3.1.
> - We added two concrete examples of the operators $A,B,\hat B$ in Section 3.1.2 to clarify their interpretation in ML settings.
>
> Detailed comments
>
> 1. *Is $\Delta J_{\mathrm{test}}=J_{\mathrm{test}}(w_0+\Delta w)-J_{\mathrm{test}}(w_0)$*
> &nbsp;
> Yes. $\Delta \mathcal{L}$ is now defined explicitly at the beginning of 3.1. (Note that in accordance with Reviewer zuhF’s suggestion, we’ve changed our notation for the loss from $J$ to $\mathcal{L}$.)
> &nbsp;
> 2. *What is the source of the noise ? Is it sampling noise, label noise, general?*
> &nbsp;
> We have clarified this in the text. The framework supports general small perturbations: sampling noise, label noise, distributional drift, etc. The only requirement is that the perturbation induces small parameter displacements for which a quadratic approximation is valid; the fluctuation formula itself does not depend on the specific noise source.
> &nbsp;
> 3. *It would help to break up Section 3 into Theorem statements.*
> &nbsp;
> We have restructured Section 3 around two new theorem statements that encapsulate the main theoretical contributions, improving clarity and flow.
> &nbsp;
> 4. *It would help to have prose at the start of Section 3, 3.1 describing what those sections do.*
> 5. *Consider separating out preliminaries for notation and results. The authors should be more explicit about things like what the noise is, what exactly $\Delta J_{\mathrm{test}}$ is, etc.*
> &nbsp;
> Done. See previous and section 3.1.
> &nbsp;
>
> *It seems like a nice paper, but I would recommend either submitting it to a different venue or writing it more in the style of ML theory papers for publication at an ML conference.*
>
> We believe these revisions address the concerns regarding style and exposition, and they have substantially improved the clarity of the manuscript.
>
> Regarding venue fit: the revised paper now follows the style and structure typical of ML theory submissions, and we believe the work sits squarely within ICML’s scope. The results provide new theoretical insights into noise-induced generalization error, and are supported by experiments on neural networks. We hope the updated presentation clarifies the relevance of our contribution to the ML community.
>
> If there are additional concrete aspects of the presentation you feel would further strengthen the paper for ICML readers, we would be very happy to incorporate them.

---

### Official Review · Reviewer_Rqbg · 2025-11-04

**Soundness:** 3
**Presentation:** 2
**Contribution:** 2
**Rating:** 4
**Confidence:** 3

**Summary:**

This paper analyzes the alignment between eigenvectors of train and test loss Hessians and its impact on generalization, extending prior works that focused mainly on eigenvalue spectra. A universal local fluctuation law is formulated to demonstrate the predictive role of eigenvector alignment for test loss. The theory is further applied to linear regression under covariate shift and multiple descent scenarios. Experiments extend the framework to neural networks, including MLPs and ResNet-20 trained on CIFAR-10, with a novel algorithm estimating eigensubspace alignment in the bulk of small eigenvalues.

**Strengths:**

- The paper provides a novel and original analysis revealing the importance of eigenvector alignment between train and test Hessians, which is an interesting and underexplored aspect in the literature.
- A new framework is introduced to evaluate the alignment between eigenvectors associated with the bulk of small eigenvalues.
- The analysis in linear regression provides concrete intuition on covariate shift and multiple descent.

**Weaknesses:**

-  Although the eigenvector alignment analysis is theoretically interesting, it is unclear how this insight could translate into practical benefits for model training, as test data are not available during training.
-  The experiments involving neural networks in Sections 3.3 and 3.4 lack clear motivation. Section 3.3 partially supports the theory in Section 3.1 but does not provide concrete insights into MLP generalization or learning dynamics. The purpose of the figures at the bottom of Figure 4 is also unclear and needs further explanation. Section 3.4 proposes an interesting scalable estimation method for bulk subspace alignment, yet the analysis does not seem to make it a central element of the argument, which leaves its importance underemphasized. Moreover, the section concludes by examining train–test Hessian misalignment under test-class imbalance, without providing additional insights into generalization.
-  The paper’s organization could be improved for readability. Key equations (e.g., (13)) and concepts (e.g., smoothing kernels in (10)) are relegated to the appendix. Terms such as “error increment” and “bias” in Figure 2(a, c) should be explicitly defined in the main text.

**Minor comments**

- The caption of Figure 4 (referring to $H_{test}$) does not match the main text (referring to $H_{train}$).

**Questions:**

-	Please refer to the points raised in the weaknesses section.
-	The developed theory resembles the Takeuchi information criterion (TIC). Could the authors discuss potential connections or differences? (see, e.g., Thomas et al., On the interplay between noise and curvature and its effect on optimization and generalization, AISTATS 2020).
-	How can the analysis be adapted to account for stochastic optimization?
-	How robust is the estimator proposed in Section 3.4? Some ablation studies on its hyperparameters would strengthen the argument.

---

> ### Author Response · Authors · 2025-11-22
> **Response 1 to Reviewer Rqbg [1/3]**
>
> **Weaknesses**
>
> 1. *Although the eigenvector alignment analysis is theoretically interesting, it is unclear how this insight could translate into practical benefits for model training, as test data are not available during training.*
> &nbsp;
> We appreciate this point, and we also appreciate the reviewer’s positive remark elsewhere that “the paper provides a novel and original analysis revealing the importance of eigenvector alignment between train and test Hessians, which is an interesting and underexplored aspect in the literature.” Our aim in this work is indeed primarily theoretical and mechanistic: given a pair of losses, the two-loss framework explains how perturbations harm generalization by decomposing the effect into spectra and eigenvector overlaps. This provides a theoretically grounded mechanism, in the same spirit as other population-level criteria in statistical learning theory such as the Takeuchi Information Criterion (TIC), which is also formulated in terms of the curvature of the population loss, rather than directly observable test data.
> &nbsp;
> Although the analysis is formulated in terms of the true test set, it remains practically valuable in several ways. For example, it provides diagnostic tools: by examining how a model’s two-loss geometry responds to different objectives one can understand why some shifts are more harmful than others. Second, while developing practical regularizers is not the focus of this paper, one could imagine estimating alignment using, for example, train-validation splits. This suggests "alignment-aware" minimization algorithms that could complement or augment standard sharpness-aware approaches. We leave this intriguing direction for further work.
> &nbsp;
> We have edited the Discussion to highlight these practical future directions.
> &nbsp;
> 2. *The experiments involving neural networks in Sections 3.3 and 3.4 lack clear motivation. Section 3.3 partially supports the theory in Section 3.1 but does not provide concrete insights into MLP generalization or learning dynamics. The purpose of the figures at the bottom of Figure 4 is also unclear and needs further explanation.*
> &nbsp;
> The purpose of Section 3.3 is methodological: to provide a controlled, nonlinear test of the local quadratic fluctuation law introduced in Section 3.1—rather than to offer new insights into MLP generalization or learning dynamics. We have edited the text at the start of Section 3.3 to make this motivation explicit.
> &nbsp;
> Although not the focus of Section 3.3, we note that Figures 6 and 7 in the Appendix do touch on these broader questions. Figure 6 shows strong alignment between train and test Hessian eigenspaces—an indicator of robustness to perturbations and therefore conceptually linked to generalization—while Figure 7 illustrates a reasonable match between empirical gradient-descent trajectories and those predicted by the quadratic theory. These supplementary results do not aim to provide deep insights into MLP generalization or learning dynamics, but they do show that the framework naturally engages with both themes.
> &nbsp;
> Finally, regarding the bottom panels of Figure 4, we have clarified in the text that these 2D loss-landscape slices are included to aid geometric intuition, rather than to serve as quantitative results.

---

> ### Author Response · Authors · 2025-11-22
> **Response 2 to Reviewer Rqbg [2/3]**
>
> 3. *Section 3.4 proposes an interesting scalable estimation method for bulk subspace alignment, yet the analysis does not seem to make it a central element of the argument, which leaves its importance underemphasized. Moreover, the section concludes by examining train-test Hessian misalignment under test-class imbalance, without providing additional insights into generalization.*
> &nbsp;
> Thank you for this thoughtful comment. While your point is well taken, we would briefly note that although the subspace-iteration method for outlier modes is standard, using it to quantify two-loss eigenvector overlaps—an aspect of loss geometry that is rarely examined—offers a novel and informative use of this algorithm in an ML setting. Nevertheless, we agree that the bulk estimator deserves clearer emphasis, and we have revised the text accordingly. In particular, the main text now explicitly references the synthetic-data experiments demonstrating the accuracy of the KPM-based bulk-overlap estimator (Appendix Figs. 8,9), and more clearly points to the CIFAR bulk-overlap results included in the supplement (now expanded with additional runs on the perturbed test set in response to the reviewer’s helpful suggestion). Although the main figure focuses on outlier overlaps for space reasons, the revised text now makes clear that the scalable bulk estimator is applied and validated as well.
> &nbsp;
> Regarding the class-imbalance experiment, our intention was not to claim new insights into generalization, but rather to provide a concrete demonstration of how a simple form of domain shift alters the two-loss geometry of a real neural network. The observed misalignment of the imbalanced test Hessian is consistent with increased sensitivity of that objective to perturbations and illustrates the type of geometric effect highlighted by our framework. We have edited the text of 3.4 to convey this more explicitly.
> &nbsp;
> 4. *The paper’s organization could be improved for readability. Key equations (e.g., (13)) and concepts (e.g., smoothing kernels in (10)) are relegated to the appendix. Terms such as “error increment” and “bias” in Figure 2(a, c) should be explicitly defined in the main text.*
> &nbsp;
> Thank you for these suggestions. We agree that the original draft needed clearer exposition, especially in Section 3.1. We have substantially revised this aspect of the paper, including the following changes:
>     1. Notation and preliminaries have been separated into a dedicated subsection at the start of Section 3.1. In particular, test error increment and bias (the baseline test loss, now notated $\mathcal{L}_0$) are explicitly defined in this subsection.
>     2. We now introduce the overlap function directly in Theorem 1 by its conceptual definition (mean squared cosine between the corresponding eigenspaces). The full detailed expression involving smoothing kernels remains in Appendix B.1 for clarity and to avoid interrupting the flow of the main text.
>     3. In Section 3.4, we now clarify that equation (10) (now equation (15)) is a general identity valid for any smoothing kernel $G$. When describing the KPM-based overlap estimator, we now explicitly state that we choose $G$ to be a Gaussian kernel for the numerical implementation.
> &nbsp;
>
> *Minor comments*
> &nbsp;
> *The caption of Figure 4 (referring to $H_{test}$) does not match the main text (referring to $H_{train}$).*
> &nbsp;
> Fixed. Thank you for bringing this to our attention.

---

> ### Author Response · Authors · 2025-11-22
> **Response 3 to Reviewer Rqbg [3/3]**
>
> **Questions**
> &nbsp;
> 1. *The developed theory resembles the Takeuchi information criterion (TIC). Could the authors discuss potential connections or differences? (see, e.g., Thomas et al., On the interplay between noise and curvature and its effect on optimization and generalization, AISTATS 2020).*
> &nbsp;
> Thank you for pointing out this important connection. We have added a brief discussion of the TIC in the introduction. The TIC analyzes the bias between population risk and empirical risk by treating the empirical loss as a sampling-perturbed version of the population loss. In that asymptotic setting, the learned parameter is close to the population minimizer, and one obtains a trace expression involving the Hessian and the Fisher information—the covariance of the gradient noise. This has the same general algebraic form as the trace expression appearing in our fluctuation law.
> &nbsp;
> Our framework generalizes this picture in two ways. First, we allow arbitrary perturbations of the training loss, not only sampling noise. Second, we work in a two-loss setting in which the train and test Hessians need not coincide. While one could in principle decompose the single-loss TIC trace into spectral and eigenvector-overlap components, this decomposition does not appear in the TIC framework. One of our primary contributions is to make this decomposition explicit and to show how overlaps between $H_{\mathrm{train}}$ and $H_{\mathrm{test}}$ eigenspaces play a central role. If the reviewer thinks a more detailed comparison is needed, we would be happy to add a short appendix note.
> &nbsp;
> 2. *How can the analysis be adapted to account for stochastic optimization?*
> &nbsp;
> We appreciate this question—Stochastic optimization fits directly into our framework. In the paper we already package the inverse Hessian and the gradient-noise covariance into a single “train covariance” $C_{\mathrm{train}}$, and all generalization effects enter only through the overlap term $\tfrac{1}{2d}\mathrm{tr}[H_{\mathrm{test}}C_{\mathrm{train}}]$. A noisy gradient-descent SDE simply provides a different $C_{\mathrm{train}}$: in the quadratic approximation its stationary covariance is the solution of a Lyapunov equation $H_{\mathrm{train}} C_{\mathrm{train}} + C_{\mathrm{train}} H_{\mathrm{train}} = Q$ (here $Q$ is the noise covariance), which plays exactly the same conceptual role as the effective covariance used in our main results.
> &nbsp;
> Importantly, this covariance is still “Hessian-filtered”—fluctuations are suppressed along high-curvature directions and amplified along shallow ones—and the expected test-loss increment decomposes in the same way into train/test spectra and eigenvector overlaps. We did not pursue this extension in the paper due to space, but nothing in the two-loss theory needs to be changed: one simply inserts the SDE’s $C_{\mathrm{train}}$ into the existing trace formula.
> &nbsp;
> These points are now discussed briefly in 3.1.1 and a short appendix note (Appendix B.2.2).
> &nbsp;
> 3. *How robust is the estimator proposed in Section 3.4? Some ablation studies on its hyperparameters would strengthen the argument.*
> &nbsp;
> The estimator in 3.4 is composed of well-studied building blocks—Chebyshev polynomial fitting and Hutchinson trace estimation—so the effect of its hyperparameters is predictable and controlled. The truncation order $K$ governs the approximation error of the Gaussian kernel, and for smooth kernels this error decays exponentially with $K$. The number of probes $P$ affects only the variance, following the standard $O(1/\sqrt{P})$ rate of Hutchinson estimators. The kernel width determines the smoothness of the overlap integrand: narrower kernels deliver higher resolution but require larger $K$, while broader kernels are easier to approximate.
> &nbsp;
> In practice we observe the expected bias-variance tradeoff, and the estimator is stable over wide ranges of $K, P$. In Appendices F.3, F.4, we have added a brief discussion of these hyperparameter effects and their theoretical scalings, along with a new supplementary figure illustrating a simple experiment varying $K,P$ on synthetic data.

---

### Meta-Review · Area_Chair_Vuaf · 2026-01-07

**Summary:**

The paper introduces a two-loss perspective on local loss geometry, arguing that generalization behavior depends not only on the spectra of the training and test Hessians but also crucially on the alignment or overlap of their eigenspaces. The core theoretical contribution is a universal local fluctuation law that expresses the expected test-loss increase under small training perturbations as a trace involving (i) the train Hessian spectrum, (ii) the test Hessian spectrum, and (iii) a new overlap functional quantifying eigenspace alignment. A related transfer law describes how overlaps transform under noise.

As an analytically tractable case, the framework is applied to ridge regression under covariate shift, using tools from operator-valued free probability to derive asymptotically exact overlap decompositions. This analysis reframes multiple descent phenomena: peaks in test error are attributed primarily to eigenspace misalignment rather than Hessian ill-conditioning alone.

On the empirical side, the paper (i) validates the fluctuation law in controlled multilayer perceptron experiments, (ii) develops scalable numerical estimators for overlap functionals (via subspace iteration for outliers and kernel polynomial methods for the spectral bulk), and (iii) applies these estimators to a ResNet-20 on CIFAR-10 to illustrate how test-set class imbalance induces train–test Hessian misalignment. The authors position overlaps as a missing but fundamental ingredient for understanding generalization, complementing spectrum-based and sharpness-based analyses.

**Reviewer Concerns:**

Initial impressions from reviewers were mixed, leaning toward a more positive assessment. More specific concerns include:

- Clarity (raised by Rqbg, G7EJ, GPhb, tXsV): Reviewers found the original manuscript hard to read, following a “physics-style”, and insufficiently explicit about notation, approximation schemes, and the meaning of operators. Key derivations (especially the fluctuation law) were viewed as opaque, with important equations and definitions pushed to the appendices. The authors substantially reorganized Section 3, added a preliminaries subsection, introduced explicit theorem statements, clarified the approximation scheme, and corrected notation. Reviewers GPhb and tXsV indicated that the revised version was significantly clearer and that their concerns were largely resolved.

- Practical relevance (raised by Rqbg): Reviewer Rqbg raised one central criticism that eigenvector alignment between train and test Hessians is not directly usable during training since the test data is unavailable. Reviewers broadly questioned whether the framework yields actionable insights beyond post-hoc explanation. The authors explicitly frame the work as mechanistic and diagnostic rather than prescriptive, drawing an analogy to TIC. They suggest possible future directions (e.g. validation splits or alignment-aware regularization) but do not provide concrete algorithms. I find this to be an acceptable response to the query.

- MLP experiments (raised by Rqbg, zuhF): Reviewers questioned the motivation of Sections 3.3 and 3.4, arguing that the MLP experiments add limited insight into learning dynamics or generalization and that the CIFAR-10 experiment design (fixed pretrained model, Hessians estimated on 5k samples) is unconventional and potentially unconvincing. The authors clarified that Section 3.3 is intended as a methodological validation of the fluctuation law, not as a study of MLP generalization. For CIFAR-10, they argue that fixing the model and varying only the test distribution isolates geometric effects. They added explanations, expanded figures, and some additional runs. The clarified motivation certainly helps here.

- Robustness (raised by Rqbg, zuhF): Reviewers asked whether the proposed bulk overlap estimator is robust to hyperparameters and sampling variability, and whether additional ablations were needed. The authors provided theoretical bias-variance arguments, added synthetic-data ablations, and discussed scaling and runtime. However, they largely deferred questions about variability of the CIFAR Hessian estimates and milder imbalance regimes to future work. Reviewer zuhF explicitly maintained concerns about the CIFAR setup, and the author’s response, while defensible, does not fully close this gap.

- Related work (raised by Rqbg, zuhF, tXsV): Reviewers requested clearer positioning relative to TIC, sharpness-aware optimization, domain-invariance methods, and prior work on eigenspace overlaps. The authors substantially expanded the related work discussion and explicitly compared their framework to TIC, emphasizing the novelty of the two-loss, overlap-based decomposition. This concern was well-addressed.

**Reviewer Scores:**

As the authors point out, there are several positive assessments, and the negative assessments came from reviewers that did not get the opportunity to respond to the author rebuttal, but might plausibly have done so, given the quality of the author responses. Most of the major concerns raised have been thoroughly addressed in the rebuttal period, and I find the resulting work to be a vast improvement that is worthy of publication.

---

### Decision · Program_Chairs · 2026-01-26

Accept (Poster)